## REPORT

# Dynamic interaction of REEP5–MFN1/2 enables mitochondrial hitchhiking on tubular ER

Shue Chen[1,2,3]*, Yang Sun[1,2]*, Yuling Qin[1,2], Lan Yang[1,2], Zhenhua Hao[4], Zhihao Xu[1,2], Mikael Björklund[2,5], Wei Liu[6], and Zhi Hong[1,2,5]

**Mitochondrial functions can be regulated by membrane contact sites with the endoplasmic reticulum (ER). These mitochondria–ER contact sites (MERCs) are functionally heterogeneous and maintained by various tethers. Here, we found that REEP5, an ER tubule-shaping protein, interacts with Mitofusins 1/2 to mediate mitochondrial distribution throughout the cytosol by a new transport mechanism, mitochondrial "hitchhiking" with tubular ER on microtubules. REEP5 depletion led to reduced tethering and increased perinuclear localization of mitochondria. Conversely, increasing REEP5 expression facilitated mitochondrial distribution throughout the cytoplasm. Rapamycin-induced irreversible REEP5–MFN1/2 interaction led to mitochondrial hyperfusion, implying that the dynamic release of mitochondria from tethering is necessary for normal mitochondrial distribution and dynamics. Functionally, disruption of MFN2–REEP5 interaction dynamics by forced dimerization or silencing REEP5 modulated the production of mitochondrial reactive oxygen species (ROS). Overall, our results indicate that dynamic REEP5–MFN1/2 interaction mediates cytosolic distribution and connectivity of the mitochondrial network by "hitchhiking" and this process regulates mitochondrial ROS, which is vital for multiple physiological functions.**

## Introduction

Mitochondria–ER contact sites (MERCs) are involved in processes essential for cellular physiology including lipid transfer, calcium transmission, and mitochondrial network dynamics (Friedman et al., 2011; Rizzuto et al., 1998; Vance, 1990). Several transmembrane protein complexes, including IP3R-Grp75-VDAC1 (Szabadkai et al., 2006), Mitofusin 2 (MFN2)-MFN2 (de Brito and Scorrano, 2008; Naon et al., 2016), Fis1-Bap31 (Iwasawa et al., 2011), and VAPB-PTPIP51 (Stoica et al., 2014), have been shown to act as tethers to mediate the interaction between ER and mitochondria. These different tethers are responsible not only for ensuring close physical proximity at MERCs but also for participating in specific functions (Prinz, 2014). For example, the Fis1-Bap31 tether is involved in apoptotic signaling (Iwasawa et al., 2011). In-depth understanding of the functional heterogeneity of tethers is largely lacking (Jing et al., 2020).

Mitofusins 1 and 2 (MFN1/2) were first identified as mitochondria-localized proteins controlling mitochondrial morphology (Rojo et al., 2002; Santel and Fuller, 2001). Later studies revealed that MFN2 tethers mitochondria to ER and this required the presence of MFN2 in the ER to form homo- or heterotypic interactions with MFN2 or MFN1 located in mitochondria, respectively (Bernard-Marissal et al., 2019; Casellas-Díaz et al., 2021; Filadi et al., 2015; Naon et al., 2016). Recently, MFN2 was shown to specifically bind and transfer phosphatidylserine (PS) from ER to mitochondria in the liver (Hernández-Alvarez et al., 2019). These discoveries illustrated the structural and functional roles of MFN2 in mediating the communication between the ER and mitochondria. However, whether there is a permanent or regulated presence of MFNs in the ER remains to be fully substantiated. Identification of new tethering factors from the ER, other than MFN2 itself, is therefore of critical importance in elucidating MERCs.

Receptor expression enhancing proteins (REEPs) are a family of ER tubular-shaping proteins previously linked to mitochondrial homeostasis. In cultured cells harboring hereditary spastic paraplegias-associated REEP1 mutations, fewer MERCs were observed compared with wild-type cells (Lim et al., 2015),

........................................................................................................................................................................

[1]Department of Neurology, the Second Affiliated Hospital of Zhejiang University, School of Medicine, Zhejiang University, Hangzhou, China; [2]Centre for Cellular Biology and Signaling, Zhejiang University-University of Edinburgh Institute, Haining, China; [3]Nuclear Organization and Gene Expression Section, Laboratory of Biochemistry and Genetics, National Institute of Diabetes and Digestive and Kidney Diseases, National Institutes of Health, Bethesda, MD, USA; [4]National Center for Children's Health, Beijing Children's Hospital, Capital Medical University, Beijing, China; [5]University of Edinburgh Medical School, Biomedical Sciences, College of Medicine & Veterinary Medicine, University of Edinburgh, Edinburgh, UK; [6]Metabolic Medicine Center, International Institutes of Medicine, the Fourth Affiliated Hospital, School of Medicine, Zhejiang University, Yiwu, China.

*S. Chen and Y. Sun contributed equally to this paper. Correspondence to Zhi Hong: zhihong@intl.zju.edu.cn

S. Chen's current affiliation is Zhejiang University, Hangzhou, China.

leading to the notion that ER-localized REEP1 shapes mitochondria. Ablation of REEP6 was reported to increase the number of mitochondria (Agrawal et al., 2017), while the loss of REEP5 was associated with the accumulation of reactive oxygen species (ROS) and impairment of mitochondrial respiration (Lee et al., 2020). The mechanisms by which REEPs confer these effects on mitochondria remain largely unknown.

In this study, we tested the hypothesis that mitochondrial MFN1 or 2 might interact with ER-localized REEPs, linking their ostensibly independent functions. By performing a small-scale protein interaction screen, we identified REEP5–MFN1/2 as a tether protein complex important in linking mitochondria and ER. Live cell imaging showed that REEP5–MFN1/2 interaction mediates subcellular distribution of mitochondria via "hitchhiking" transport on ER tubules leading to normal cytosolic distribution and connectivity of the mitochondrial network. We found that the REEP5–MFN1/2 interaction dynamics is important for maintaining the balance between mitochondrial hitchhiking, for the release from ER tubules, and for maintaining mitochondrial oxidative homeostasis. This study thus uncovers a previously unrecognized REEP5–MFN1/2 tether involved in mitochondrial hitchhiking transport, as well as the regulation of mitochondrial ROS in human cells under physiological conditions.

## Results and discussion
### REEP5 interacts with MFN1/2 at ER–mitochondria contacts
To examine which proteins exhibit direct and physical interaction(s) with Mitofusins (MFNs) to form a tether between the ER and mitochondria, we conducted a two-step candidate screen with REEPs or Reticulons, two ER-resident protein families that form membrane-bound hairpin structures that shape tubular ER. To conduct the screen, Flag-MFN1/2 and a panel of recombinant REEP proteins, including Myc-REEP1/2/3/4/5/6, and Reticulon proteins, and Rtn1/2/3/4-Myc were co-transfected into HEK239T cells. Co-immunoprecipitation (Co-IP) using Myc-antibody revealed that REEP5 displayed the strongest interaction with MFN1 and MFN2 (Fig. 1, A and B), while Rtn3 displayed a weak interaction and with MFN2 only (Fig. S1, A and B). Since disruption of tether integrity usually alters the mitochondrial dynamics (Hemel et al., 2021), we generated individual knockdown (KD) cellular models for REEP5 and Rtn3 by transient transfection with siRNAs in U2OS cells and examined changes in mitochondrial morphology. The results (Fig. S1 C) showed that the most dramatic changes occurred in REEP5 KD cells, as reflected by the loss of normal mitochondrial distribution and by the increased perinuclear clustering. IP assays using MFN1/2 antibodies showed that endogenous MFN1/2 consistently interacts with REEP5 (Fig. 1 C). To confirm the REEP5–MFN1/2 interaction, we expressed an MFN2 mutant (MFN2$^{ActA}$) with restricted expression to the mitochondrial surface (Casellas-Díaz et al., 2021; de Brito and Scorrano, 2008). IP analysis again showed that REEP5 interacts with mitochondria-localized MFN2 (Fig. 1 D).

Using domain structure truncation mutants of MFNs and REEP5 (Fig. 1 E), we determined that the C-terminal cytoplasmic domain of REEP5 (Fig. 1 F) and the C-terminal cytoplasmic domains of MFN1/2 (Fig. 1 G) were responsible for the interactions, suggesting that these proteins could form a tether at membrane contact sites. In vitro GST pull-down assays using purified His-MBP-REEP5 and GST-tagged C-terminal cytoplasmic domains of MFN1/2 further showed that the C-terminus of both MFNs coprecipitated with REEP5, supporting a direct interaction (Fig. 1, H and I).

Next, we examined whether REEP5 and MFNs co-localized at MERCs. However, immunofluorescence (IF) signals for both REEP5 and MFNs were obscured by the closely intermingled networks of ER and mitochondria, as well as the broad distributions of REEP5 and MFN1/2 in these subcellular structures. To circumvent this problem, we employed hypotonic swelling to induce large intracellular vesicles (LICVs). This method has emerged as an effective tool for observing organellar membrane localization of tether proteins at the membrane contact sites (King et al., 2020). COS7 cells transfected with mEmerald-REEP5 and HaloTag-MFN1/2 were treated with a hypotonic medium to create REEP5-positive ER-LICVs and MFN1/2-positive Mito-LICVs. To avoid inducing an abnormal mitochondrial phenotype due to MFN1/2 overexpression (Huang et al., 2007; Santel and Fuller, 2001), we transfected cells with various amounts of MFN1/2 plasmids and selected the minimal amount of ectopically expressed MFN2, which is ∼20% of the endogenous protein, and retained normal mitochondrial morphology during imaging (Fig. S1 D). While few REEP5-positive ER-LICVs could be observed in the majority of cells (Fig. 2 A, first and third rows, 15/19 cells), the ER generally retained a tubular morphology under REEP5 overexpression (Fig. 2 A, second and fourth rows), consistent with a structural role for REEP5 in maintaining ER tubules (Hu et al., 2008; Wang et al., 2021). Nevertheless, the ends of REEP5-positive ER tubules appeared tightly connected to Mito-LICVs, with enrichment for mEmerald-REEP5 and HaloTag-MFN1/2 signals at those membrane contact sites (Fig. 2 A, second and fourth rows, and Video 1). There was also an enrichment for both proteins at the contact sites between Mito-LICVs and ER-LICVs (Fig. 2 A, first and third rows, and Video 2).

We then reasoned that the mobility of the REEP5–MFN1/2 protein pair at the membrane contact sites is likely retarded by their interaction. To examine this possibility, we used fluorescence loss in photobleaching (FLIP) to eliminate the mEmerald-REEP5 signal in the LICV membranes away from contact sites in MFN1&2 double knockout (DKO) MEFs and found that the REEP5 signals remained at the membrane contact sites in the presence of MFN2, while they quickly disappeared in the absence of MFN2 (Fig. 2 B and Video 3). Meanwhile, mEmerald-Sec61β exhibited no sensitivity to MFN2 and quickly dispersed at the contact sites upon photobleaching (Fig. 2 B and Video 3). These results suggest that REEP5 in the ER interacts and co-localizes with mitochondrial MFN1/2 at the membrane contact sites.

Given the direct contact nature of the REEP5–MFN1/2 interaction, one might expect that the free C-terminal tail of REEP5 is sufficient to support its recruitment to mitochondria through binding to MFN1/2. Expression of the monomeric REEP5 C-terminus in MFN1&2 DKO MEFs showed a detectable but moderate

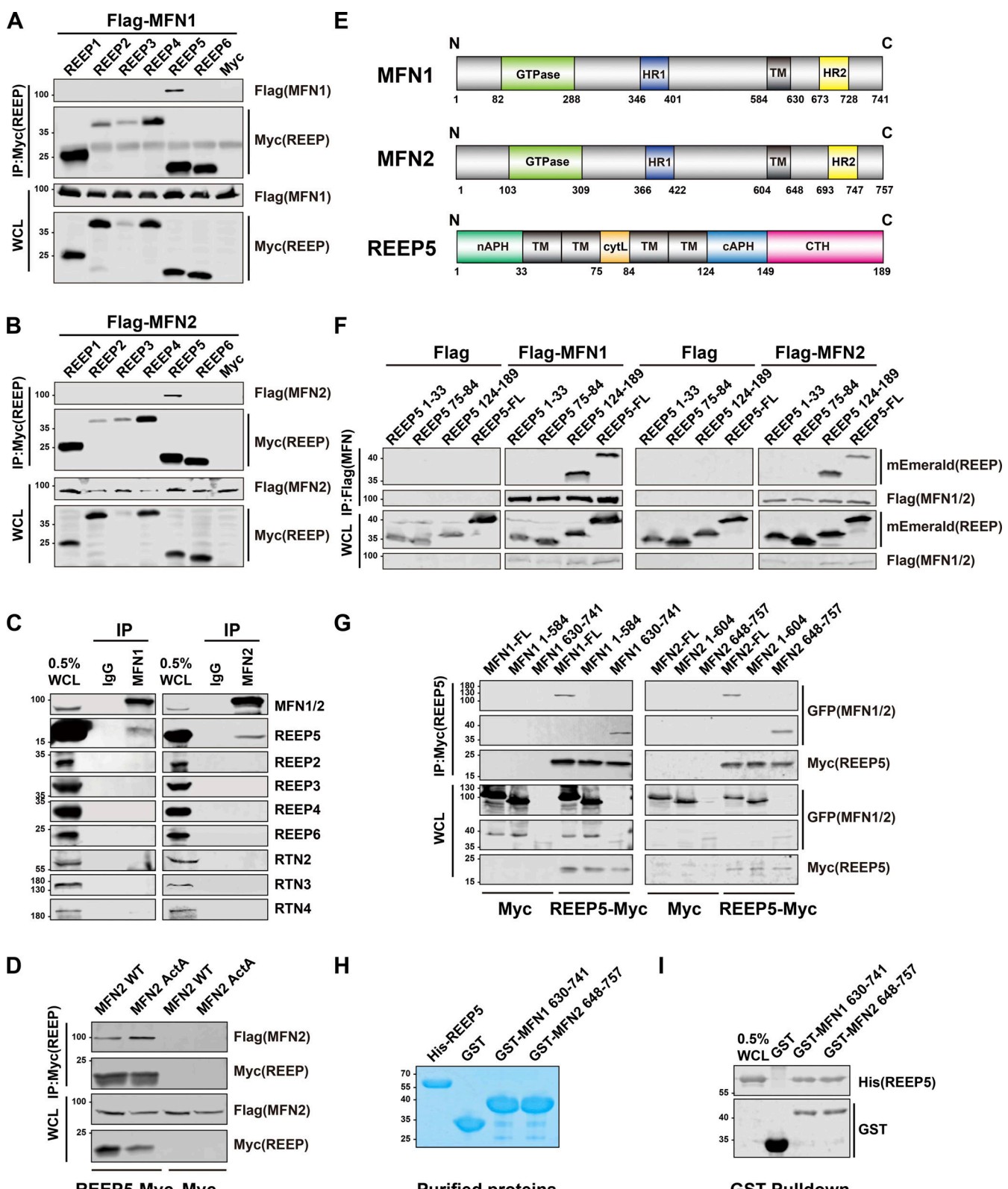

Figure 1. **REEP5 interacts with MFN1/2 through C-terminal cytosolic domains.** All co-immunoprecipitation (Co-IP) experiments were done in HEK293T cells. **(A and B)** Co-IP analysis of Myc-tagged REEP 1–6 with Flag-MFN1/2. **(C)** Co-IP analysis of endogenous MFN1/2 with REEPs and Rtns. **(D)** Co-IP analysis of Flag-MFN2 wild type or Flag-MFN2[ActA] with REEP5-Myc. **(E)** Schematic diagram of human REEP5 as previously reported (Lee et al., 2020; Xiang et al., 2023), MFN1 according to UniProt, and MFN2 according to Filadi et al. (2018). HR, coiled-coiled heptad repeat domain; TM, transmembrane; APH, amphipathic helix; cytL, cytosolic loop; CTH, C-terminal helix. **(F and G)** Co-IP analysis of mEmerald-tagged REEP5 FL or truncation mutants with Flag-MFN1/2, and GFP-tagged MFN1/2 FL or truncation mutants with REEP5-Myc. FL, full length. Numbers indicate amino acids in each construct. **(H)** Purified His-REEP5, GST-MFN1/2 C-terminal cytosolic domains, and GST were analyzed by Coomassie staining. **(I)** GST pull-down of GST-MFN1/2 C-terminal cytosolic domains with His-REEP5. Source data are available for this figure: SourceData F1.

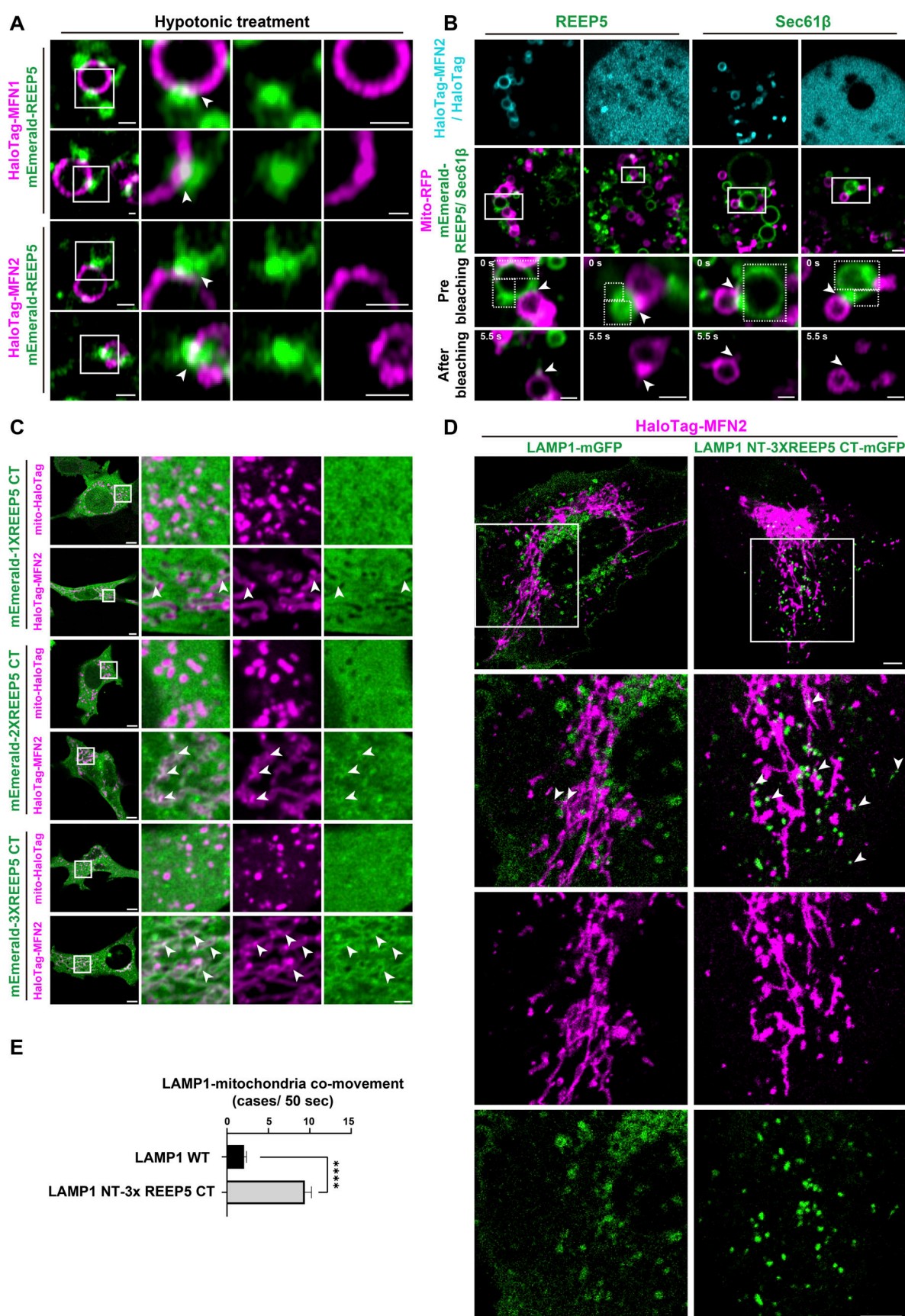

**Figure 2. REEP5-MFNs serve as molecular tethers at MERCs. (A)** Formation of ER–mitochondria contact sites under hypotonic conditions (also see Videos 1 and 2). COS-7 cells were co-transfected with HaloTag-MFN1/2 and mEmerald-REEP5. White arrowheads, contact sites. Bars: 1 µm. **(B)** ER–mitochondria LICV

contact sites before and after photobleaching (also see Video 3). MFN1&2 DKO MEFs were transfected with mEmerald-REEP5/Sec61β and HaloTag-MFN2/HaloTag. Mitochondria were labeled with mito-RFP. Dash-line boxes, bleached regions. White arrowheads, contact sites. Bars: 2 μm. A total of six cells from three independent experiments were analyzed for each group. **(C)** Confocal images of MFN1&2 DKO MEFs co-transfected with mEmerald-REEP5 C-terminus (CT) and mito-HaloTag or HaloTag-MFN2 (also see Video 4). White arrowheads, REEP5 CT recruitment onto MFN2-mitochondria. Bars: 5 μm (whole cell view, left); 2 μm (zoomed-in images). **(D)** Confocal images of U2OS cells co-transfected with HaloTag-MFN2 and mGFP tagged LAMP1 wt or chimera (LAMP1 NT-3XREEP5 CT). White arrowheads, co-movement events (also see Video 5). Bars: 5 μm. **(E)** Comparison of LAMP1–mitochondria co-movement cases within 50 s between LAMP1 wt or LAMP1 chimeric mutant expressing cells. Values shown are means ± SEM (n = 15).

enrichment of REEP5 signal in the presence of MFN2 (Fig. 2 C, second row). However, two or three tandem copies of the C-terminus exhibited significant recruitment onto mitochondria when MFN2 was present (Fig. 2 C, fourth and sixth rows, and Video 4). Importantly, these structures are negative for the ER marker-KDEL (Video 4), reflective of mitochondria-recruited REEP5 C-terminus protein.

To further validate the tethering function of the REEP5 C-terminus, we replaced the cytoplasmic portion of LAMP1 with three copies of the REEP5 C-terminus. Expression of the chimera but not wild-type LAMP1 together with MFN2 indeed increased the contact frequency of mitochondria with lysosomes (Fig. 2, D and E; and Video 5), consistent with the idea that REEP5 and MFN2 form a tethering complex.

### REEP5–MFN1/2 interaction mediates mitochondrial hitchhiking on ER tubules

Tubular ER decorated with tubule-shaping proteins, including REEP5, can move along microtubules (MTs) (Friedman et al., 2010; Goyal and Blackstone, 2013; Zheng et al., 2022). The tethering of mitochondria to tubular ER by REEP5–MFN1/2 interaction led us to test whether tethered mitochondria could be co-transported with the tubular ER along MT, an organelle transport mechanism previously described as "hitchhiking" (Guimaraes et al., 2015; Salogiannis et al., 2016). To avoid mitochondrial hyperfusion caused by REEP5–MFN1/2 overexpression, we established a U2OS cell line stably expressing low levels of MFN2 (Fig. S1 D) and REEP5. Using high-resolution live-cell grazing incidence structured imaging microscopy (GI-SIM), we found that MFN2-positive mitochondria could indeed move with REEP5-decorated ER tubules (see Fig. 3, A' and A''; and Video 6). Specifically, an individual mitochondrion appeared to move directionally with the tubular ER. Upon its release from ER contact, the unidirectional movement of the mitochondrion ceased immediately (Fig. 3 A'' and Video 6). The released mitochondria could subsequently attach to another tubular ER structure and its co-movement was reinitiated, a process we termed a "catch-release-recatch cycle" (CRRC) (Fig. 3 A'''' and Video 6). A prominent feature of CRRC is that mitochondria lose their directional movement when released from a tubular ER but regain directionality upon reattachment.

A recent report suggested that the ER is involved in lysosome translocation to the cell periphery via kinesin recruitment to the lysosomal membrane during membrane contact (Raiborg et al., 2015). It was shown that lysosomal directional motility is accelerated once leaving the ER. We did not observe any significant acceleration of mitochondrial movement upon release from ER tubules (Fig. 3, A'' and A''''; and Video 6) and we thus excluded the possibility of a similar "loading motor" role for REEP5–

MFN1/2 interaction. In fact, fewer "hitchhiking" events were detected in control cells co-expressing Sec61β instead of REEP5 in GI-SIM analyses (Fig. 3 B and Video 7), further supporting a role for REEP5 in ER–mitochondrial co-movement. MT depolymerization by nocodazole blocked this co-movement (Fig. 3 C), illustrating the microtubule dependency of the hitchhiking process.

To further confirm that REEP5 and MFN1/2 could function coordinately as a tether to mediate mitochondrial hitchhiking on ER, we analyzed the movement of mitochondria labeled with mito-RFP in WT or MFN1&2 DKO MEFs. Due to the extensive mitochondrial fragmentation in DKO cells, the mitochondrial movement could not be directly compared between MFN1&2 DKO and WT MEFs. Instead, we analyzed the effects of mEmerald-REEP5 overexpression on mitochondrial movement in MFN1&2 DKO or WT MEFs. MEFs expressing mEmerald or mEmerald-REEP5 were mixed so that the image could be captured simultaneously to avoid heterogeneity in different imaging sessions (Fig. 3 D). The results showed that REEP5 expression was sufficient to increase the number of mitochondria exhibiting directional movement in cells with functional MFN1/2, whereas no increase in the directional movement was detected in MFN1&2 DKO MEFs (Fig. 3, E–G; and Videos 8 and 9). These results indicated that REEP5–MFN1/2 interaction played a noticeable role in mitochondrial hitchhiking.

### Cytoplasmic distribution of mitochondria requires appropriate levels and reversibility of REEP5 and MFN1/2 interaction

In addition to the increased directional movement in cells expressing REEP5, we also noticed enhanced signal intensity of individual mitochondrions in REEP5-expressed WT MEFs. We therefore hypothesized that REEP5 overexpression might lead to mitochondrial hyperfusion by prolonging mitochondrial tethering to ER. Indeed, mitochondrial hyperfusion was significantly elevated as quantified by an increase in organelle aspect ratio (Song et al., 2015) in REEP5 expressing WT MEFs, while no difference in MFN1&2 DKO MEFs was observed (Fig. 4 A). We then utilized a photoswitchable mito-Dendra2 reporter to confirm the occurrence of mitochondrial hyperfusion in REEP5-overexpression MEFs. Upon randomly switching one mitochondrial subpopulation from green to magenta, we found that REEP5 expression resulted in much faster diffusion of the magenta signal throughout the whole mitochondrial network in WT MEFs, whereas no difference in the diffusion rate of the mitochondrial signal was observed between REEP5-expressing and the BFP vector control in DKO cells (Fig. 4 B). These results implied that REEP5 expression increased directional movement and consequentially mitochondrial hyperfusion. Combined with our observation of the CRRC, it appears that hitchhiking is a

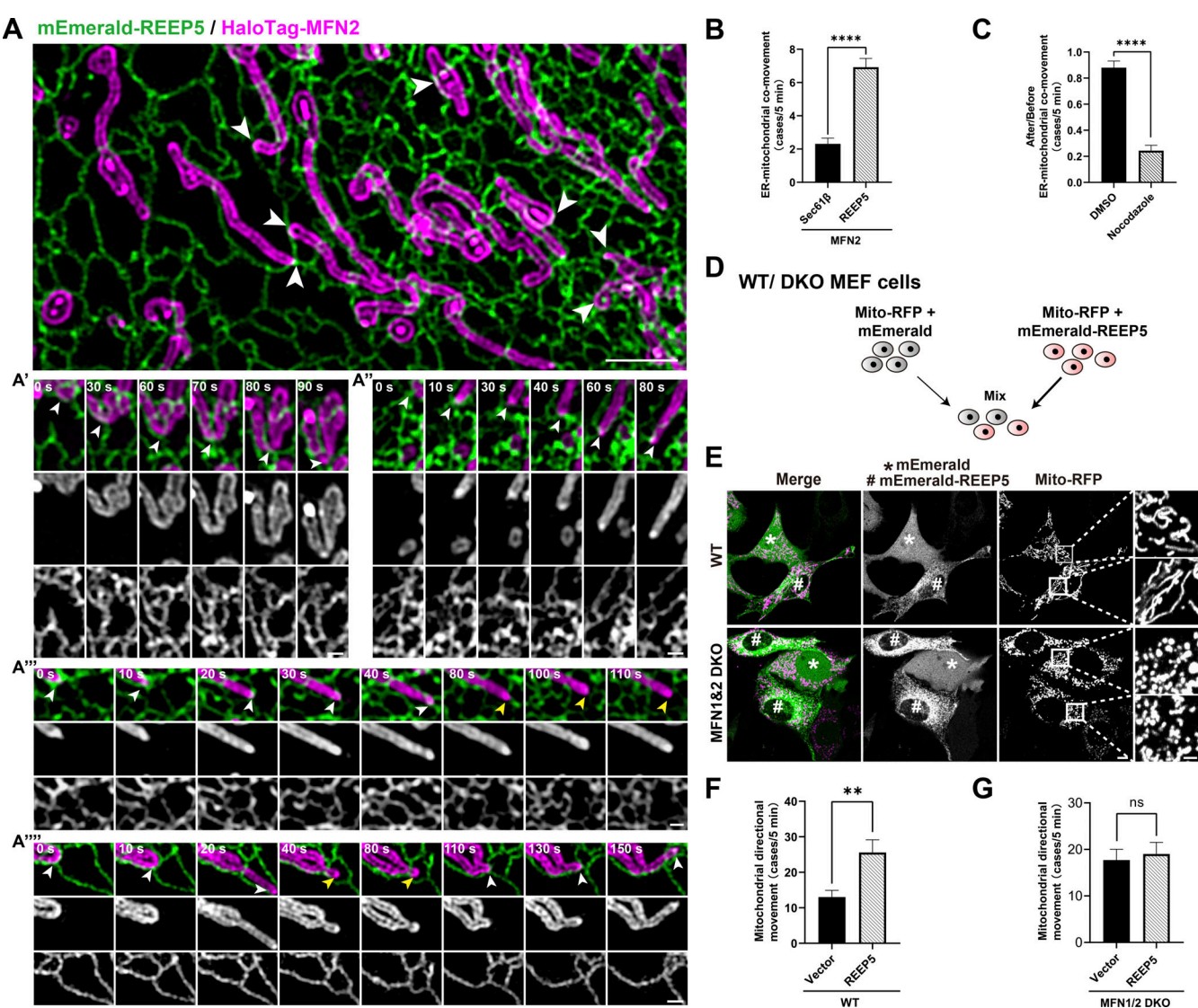

Figure 3. **REEP5 and MFN1/2 tethering facilitates mitochondrial hitchhiking on tubular ER. (A)** Top**:** GI-SIM image of ER–mitochondria contacts in U2OS cell expressing REEP5 and MFN2 (also see Video 6). Bar: 2 μm. **(A′–A″″)** Time-lapse images of ER–mitochondria dynamic co-movements in four patterns: (A′) middle part or (A″) tip of mitochondria co-moved with tubular ER; (A‴) mitochondria comoved with tubular ER, stopped moving upon dissociation; (A″″) catch-release-recatch cycle (CRRC). White arrowheads, ER-mitochondria contact sites; yellow arrowheads, ER-mitochondria dissociation. Bar: 0.5 μm. **(B)** Comparison of ER-mitochondria co-movement cases within 5 min between REEP5+MFN2 and Sec61β+MFN2 co-expressing cells (also see Video 7). **(C)** Effect of nocodazole treatment (5 μM, 15 min) on ER-mitochondria co-movement cases within 5 min in REEP5+MFN2 co-expressing cells. DMSO as control. **(D)** Schematic workflow of E. **(E)** Whole-cell-view of WT (first row) or DKO (second row) MEFs expressing mEmerald-REEP5 (labeled by #) or mEmerald (labeled by *). Bar: 20 μm. Enlarged images on the right column show the representative mitochondrial morphology from the white box as indicated. Bar: 5 μm. **(F and G)** Comparison of mitochondrial directional movement cases within 5 min between mEmerald-REEP5 or mEmerald expressing MEFs (also see Videos 8 and 9). Values shown are means ± SEM, n = 13 for B; n = 8 for C; n = 7 for F and G.

dynamic process, which requires coordination of mitochondrial directional movement to a cytoplasmic target site and appropriate timing of release (collectively termed, hitchhiking dynamics). Therefore, we further examined how REEP5–MFN1/2–associated mitochondrial attachment, "hitchhiking" on ER, and detachment are regulated.

To directly test the premise above, we adopted three strategies. First, we knocked down REEP5 to reduce the amount of REEP5 available to interact with MFN1/2. REEP5 KD leads to the perinuclear distribution of mitochondria (Fig. 4, C and D), while no noticeable changes in global ER morphology were observed

(Fig. S2 A). To test the specificity of the REEP5-MFNs interaction, we derived MFNs-binding-deficient REEP5 mutants. Among the six REEP family members, only REEP5 exhibited specific interaction with MFNs via its C-terminus (Fig. 1 F). We generated two chimeric constructs: (1) REEP5 N-terminus + REEP6 C-terminus and (2) REEP6 N-terminus + REEP5 C-terminus (Fig. S2 B). Surprisingly, both chimeras were able to interact with MFN2 (Fig. S2 C). One possibility is that the N-terminal portion of REEP6 can potentially mask or interfere with its C-terminal domain function, preventing its interaction with MFN1/2. To further examine the selectivity of REEP5 by MFN1/2, we mutated

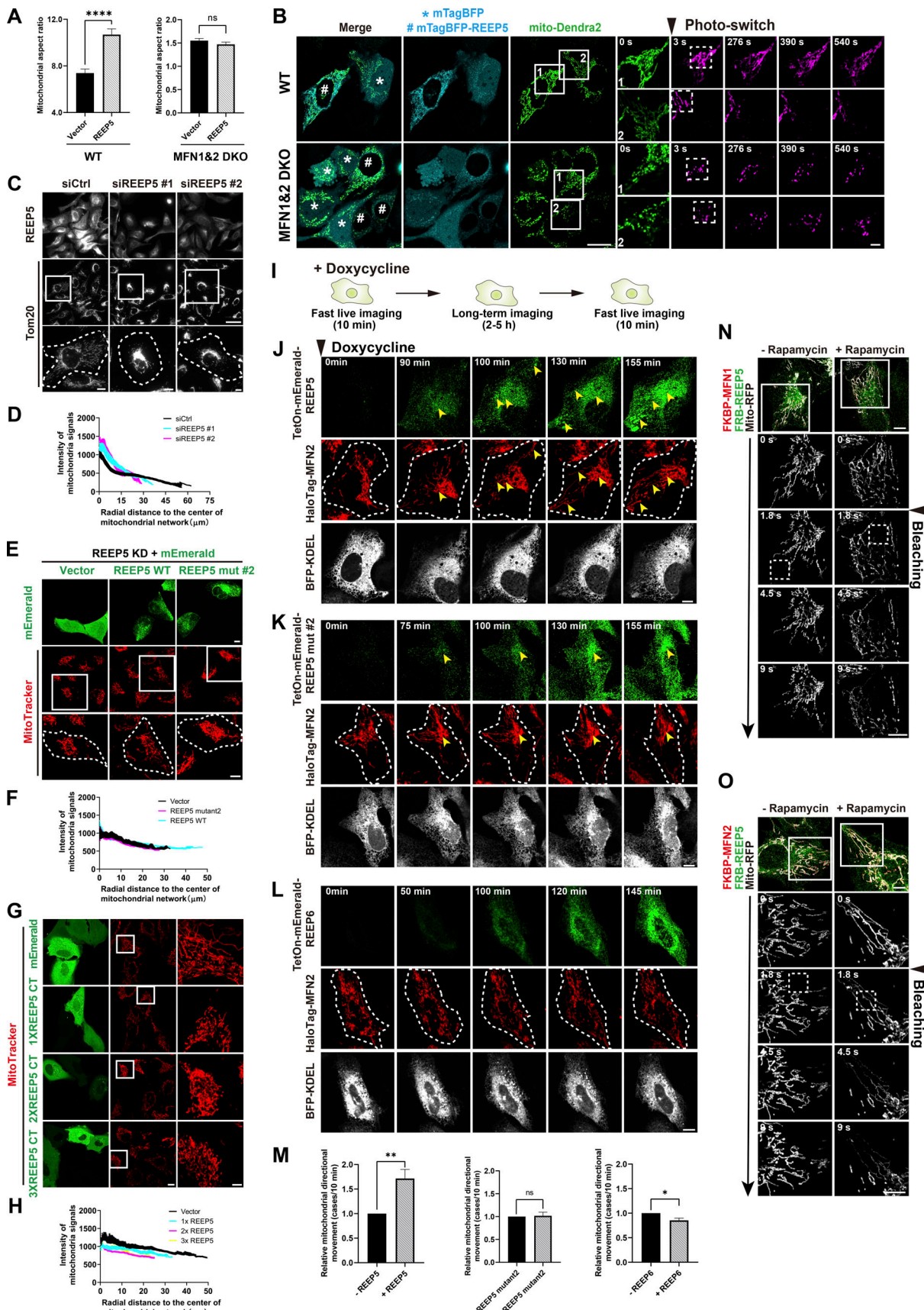

Figure 4. **REEP5–MFN1/2 interaction is involved in mitochondrial distribution. (A)** Comparison of mitochondrial aspect ratio between mEmerald-REEP5 or mEmerald expressing MEFs. **(B)** Time-lapse images of mitochondrial dynamics in MEFs co-transfected with mito-Dendra2 and mTagBFP-REEP5 or

mTagBFP. mito-Dendra2 was photo-switched by illuminating the indicated region once. Bars: 20 μm (whole cell view, left); 5 μm (zoomed-in images). Numbered boxes, enlarged area; dash-line boxes, illuminated regions. **(C and D)** Mitochondrial distribution upon REEP5 knockdown. Endogenous mitochondria and REEP5 were immunostained with anti-Tom20 and anti-REEP5 in U2OS cells in C. The radial distance to the center of mitochondrial network was quantified in D. Bars: 20 μm (whole cell view); 5 μm (zoomed-in images). **(E and F)** Comparison of mitochondrial distribution in U2OS cells transiently expressing mEmerald, siRNA-resistant mEmerald tagged REEP5 wt or mutant #2 after REEP5 depletion in E. MitoTracker labeled mitochondria. The radial distance to the center of the mitochondrial network was quantified in F. Bars: 10 μm (whole cell view); 5 μm (zoomed-in images). **(G and H)** Mitochondrial morphology from cells expressing mEmerald or mEmerald-REEP5 C-terminus (CT). MitoTracker labeled mitochondria. The radial distance to the center of mitochondrial network was quantified in H. Bars: 10 μm (whole cell view); 5 μm (zoomed-in images). **(I)** Schematic imaging workflow of doxycycline-induced REEPs expression on mitochondrial dynamics. **(J–L)** Long-term time-lapse images of ER-mitochondria movements as the accumulation of REEP5 wt (J) or REEP5 mutant #2 (K) or REEP6 (L) over time in U2OS cells. 200 nM doxycycline and 5 μM Trolox were added before imaging. Yellow arrowheads indicate the dynamic correlation between MFN2 and REEP5. Bars: 10 μm. **(M)** Quantification of mitochondrial directional movement within 10 min fast live imaging window, as indicated in I, before and after REEPs expression (also see Videos 10, 11, and 12). **(N and O)** U2OS cells were transfected with FKBP-HaloTag-MFN1/2, FRB-mEmerald-REEP5 and mito-RFP. Cells were treated with or without 20 nM rapamycin, subregions of mito-RFP network within the dash-line boxes were repeatedly photobleached (also see Videos 15 and 16). Dash-line boxes, illuminated regions. Bars: 10 μm. Values shown are means ± SEM, n = 35 for A; n = 15 for D; n = 15 for F; n = 17 for H; n = 10 for M.

REEP5 into a more REEP1-4-like sequence. We designed several point mutations intended to disrupt cAPH and CTH, the two domains in the C-terminus of REEP5 (Fig. S2 D). Mutations in the CTH domain (REEP5 mut#1) affected the interaction substantially, whereas additional mutations in the cAPH domain (REEP5 mut#2) further abolished its interaction with MFN2 (Fig. S2 E). Both mutants displayed normal ER patterns consistent with unperturbed wild-type cells, suggesting that they still maintain the ER-tubule forming ability (Fig. S2 F). Wild-type REEP5 markedly reversed mitochondrial collapse caused by REEP5 depletion, but not the non-binding REEP5 mut#2 (Fig. 4, E and F). We also overexpressed the REEP5 C-terminus in 293T cells to compete with the endogenous REEP5–MFN2 interaction (Fig. S2 G). Ectopically expressed REEP5 C-terminus in U2OS cells caused the collapse of mitochondria into perinuclear regions (Fig. 4, G and H). All these data suggested that REEP5–MFN2 interaction is essential for the normally dispersed cytoplasmic distribution of mitochondria.

Next, we increased REEP5 availability using doxycycline-inducible expression and tracked hitchhiking dynamics (Fig. 4 I). The results showed that MFN2-labeled mitochondria gradually translocated to the cell periphery with increasing amplitudes of REEP5 (Fig. 4 J). Quantification of the live imaging data revealed that wild-type REEP5 expression stimulated directional co-movement of mitochondria with ER (Fig. 4 M and Video 10). In contrast, either REEP5 mut#2 or REEP6 expression in parallel did not produce any increase in directional co-movement (Fig. 4, K–M; and Videos 11 and 12). These results indicated that wild-type REEP5 was specifically responsible for directional ER-mitochondrial co-movement. Moreover, mitochondrial distribution was highly correlated with ER as wild-type REEP5 accumulated (Fig. 4 J), while this correlation decreased under REEP5 mut#2 expression, and completely abolished under REEP6 overexpression (Fig. 4, K and L), further supporting the specificity of REEP5 in this ER–mitochondria correlation.

Finally, we determined whether dissociation of the REEP5–MFN1/2 complex contributes to mitochondrial detethering from tubular ER during hitchhiking transport. We used rapamycin-induced irreversible dimerization to convert the dynamic interaction between MFNs and REEP5 into a static interaction. As a result, the mitochondrial MFN1/2 signal exhibited

substantially elevated overlap with that of REEP5 in ER networks (Fig. S2 H; and Videos 13 and 14). Photobleaching mito-RFP in one region resulted in the rapid loss of mito-RFP fluorescence throughout the whole network (Fig. 4, N and O; and Videos 15 and 16), suggesting that irreversible anchoring due to rapamycin exposure induced mitochondrial hyperfusion.

Collectively, these data indicated that either reducing or increasing tethering disrupts mitochondrial distribution and directional movement with ER, further supporting that switching between binding and release is essential for hitchhiking dynamics.

## REEP5 and MFN1/2 interaction is required for the maintenance of mitochondrial ROS homeostasis

Subcellular distribution of mitochondria has been linked to several cellular functions, notably ROS homeostasis, which is essential for cell survival and stress responses (Frederick and Shaw, 2007; Park et al., 2011). Since we found that REEP5–MFN1/2–mediated hitchhiking was responsible for mitochondrial distribution and connectivity and previous reports showed that REEP5 depletion increased cytosolic ROS in cultured mouse neonatal cardiac myocytes (CMNCs) (Lee et al., 2020), we examined the role of REEP5–MFN1/2 interaction in mitochondrial ROS homeostasis. We quantified mitochondrial ROS dynamics in U2OS cells stably expressing a mitochondrial ROS biosensor, mito-roGFP2. As a control, quantitative live imaging showed that incubation with $H_2O_2$ increased while dithiothreitol (DTT) lowered ROS, indicating that the mito-roGFP2 reporter responds to changes in mitochondrial ROS in these cells (Fig. 5 A). We then knocked down REEP5 in mito-roGFP2-expressing U2OS cells. Internal control cells were generated by transfecting REEP5 siRNA plus HaloTag-siRNA-resistant REEP5 (HaloTag-REEP5 si[r]) into the roGFP2 cell line (Fig. 5 B). The internal control cells were mixed with either Control KD or REEP5 KD cells and used for the normalization of the ROS signal. This allowed us to avoid the heterogeneity of ROS levels and minimize variability between imaging sessions. We found that mitochondrial ROS was significantly higher in REEP5 KD cells than in Control KD cells (Fig. 5, B and C), consistent with the observations of elevated cytosolic ROS in REEP5 KD cells (Lee et al., 2020). Meanwhile, U2OS cells stably expressing mito-HyPer7, another ROS probe (Pak et al., 2020), responded to $H_2O_2$ and

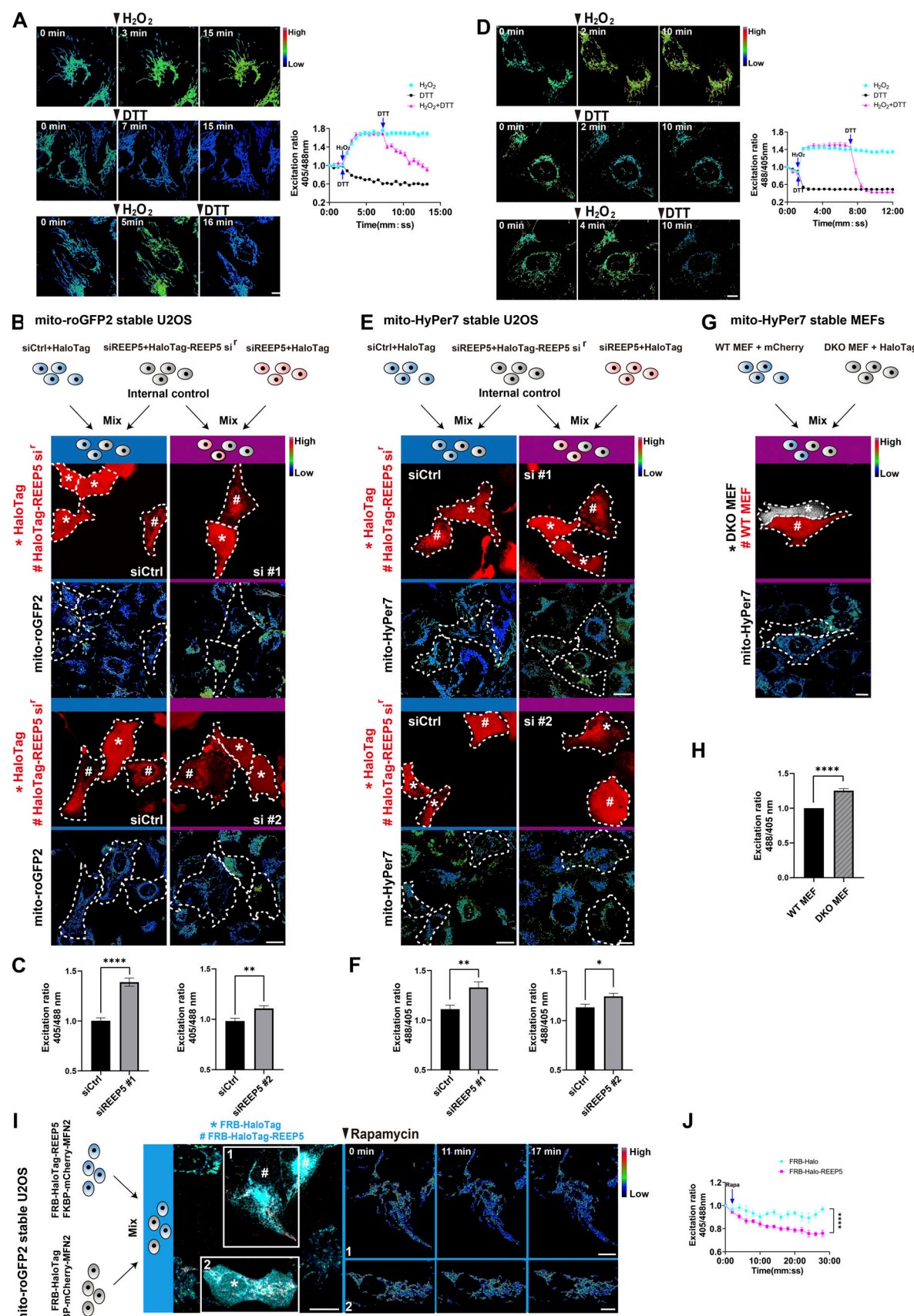

Figure 5. **REEP5 and MFN1/2 interaction regulates mitochondrial ROS level.** The fluorescence ratio of 405 and 488 nm excitation was shown by heatmap. **(A)** Time course of roGFP2 fluorescence excitation (405/488 nm) ratiometric images (R405/488) in mitochondria of mito-roGFP2 stable U2OS cells after the

addition of 200 µM H₂O₂, 1 mM DTT, or H₂O₂ + DTT for the indicated time. The ratiometric from indicated time points were calculated and shown in the graph. Bar: 10 µm. **(B)** Schematic workflow showing the internal control cells (labeled by #) were respectively mixed with either Ctrl siRNA (labeled by *) or REEP5 siRNA (labeled by *) transfected cells. The corresponding ratiometric images of mito-roGFP2 are shown below. Bar: 20 µm. **(C)** Quantification of mitochondrial ROS level from B. **(D)** Time course of HyPer7 fluorescence excitation (488/405 nm) ratiometric images (R488/405) of mito-HyPer7 stable U2OS cells exposed to 4 µM H2O2, 20 mM DTT, or H2O2 + DTT for the indicated time. Bar: 10 µm. The ratiometric changes overtime were calculated and shown in the graph. **(E)** Same as B, images of HyPer7 fluorescence excitation (488/405 nm) ratio. Bars: 20 µm. **(F)** Quantification of mitochondrial ROS level from E. **(G)** Wild-type and MFN1&2 DKO MEFs stably expressing mito-HyPer7 were mixed as in the schematic workflow, images of mitochondrial ROS are shown below. Bar: 20 µm. **(H)** Quantification of mitochondrial ROS level from G. **(I)** Schematic workflow and images of mitochondrial ROS changes upon rapamycin-induced REEP5-MFN2 binding. 40 nM Rapamycin was added at 0 min. Bars: 20 µm (whole cell view); 10 µm (zoomed-in images). **(J)** Quantification of mitochondrial ROS level from I. Values shown are means ± SEM, n = 6 for A; n = 7 for D; n = 25 for C; n = 22 for F; n = 24 for H; n = 7 for J.

DTT, similar to the mito-roGFP2 probe (Fig. 5 D). REEP5 KD in this cell line also significantly increased mitochondrial ROS levels (Fig. 5, E and F). Moreover, we established MEF cells stably expressing mito-HyPer7, and their quantification showed that MFN1&2 DKO leads to increased mitochondrial ROS (Fig. 5, G and H). ER is also a prevalent source of intracellular ROS (Hwang et al., 1992; Yoboue et al., 2018). To determine whether the increased ROS production originates from ER, we established a U2OS cell line stably expressing ER-TriPer (Fig. S3 A), a biosensor for ER ROS (Melo et al., 2017). Quantification of the ROS imaging showed no significant change in ER ROS level (Fig. S3, B and C). This result provided additional evidence that the increased ROS is produced in the mitochondria.

We then induced irreversible REEP5–MFN2 interaction in mito-roGFP2-expressing U2OS cells. Mitochondrial ROS levels decreased following the addition of rapamycin, especially in those mitochondrial tubules extending from the perinuclear mitochondrial cluster along REEP5-labeled ER tubules (Fig. 5, I and J). Quantitative imaging of ROS indicated that REEP5–MFN2–mediated mitochondrial tethering to ER could alleviate ROS accumulation in the mitochondria (Fig. 5 J). Taken together, these data suggest that a balanced dynamics of REEP5–MFN2 interaction is crucial for mitochondrial ROS homeostasis.

In summary, our study identifies the REEP5–MFN1/2 complex as an ER–mitochondria tether and reveals its functions in regulating cytosolic distribution and connectivity between mitochondria and ER. We show that this interaction is prevalent in the maintenance of ROS homeostasis in cells. Previous reports suggested that various membraneous and non-membraneous organelles are transported indirectly via hitchhiking along microtubules (Baumann et al., 2012; Guimaraes et al., 2015; Liao et al., 2019; Pohlmann et al., 2015; Salogiannis et al., 2016; Salogiannis and Reck-Peterson, 2017). The REEP5–MFN1/2 interaction enables mitochondrial hitchhiking on tubular ER, a novel mechanism that has not been demonstrated before. We dissected and interrogated the REEP5–MFN1/2 interactions by manipulating REEP5 levels, irreversible dimerization, interaction disruption, and mistargeting this interaction from ER–mitochondria to lysosome–mitochondria. It is noticeable in endogenous IP that if there is an upshifted mobilities for MFN1/2, it is possible that REEP5 binds with a subpopulation of posttranslationally modified MFN1/2, which warrants future investigations. From these evidences, we believe that a dynamic interaction between REEP5 and MFN1/2, assuming the "catch-release-recatch cycle," is critical for mitochondria hitchhiking. Our results thus suggest a model in which both REEP5 protein

levels and tether complex disassembly are tightly orchestrated for mitochondrial distribution and network connectivity. Our findings provide a new perspective to the study of organelle transport as future research should not only study the tethering of organelles but also their dissociation. In our study, the intrinsic nature of the interaction between the tethers renders the interaction dynamic, which is a key mechanism for achieving a uniform distribution of mitochondria. This raises the question of whether there are distinct and specific mechanisms to mediate the release of "hitchhiking" organelles.

The increased mitochondrial ROS in REEP5-depleted cells confirmed that REEP5 and ER tethering are important in maintaining mitochondrial ROS homeostasis. By inducing the dimerization and enhanced interaction of FKBP–MFN2 and FRB–REEP5, we observed a decrease in mitochondrial ROS. Although the increased interaction of an artificial dimer does not reflect physiological interaction in vivo, our data clearly show that mitochondrial ROS levels depend on the association between ER and mitochondria at MERCs. Given that different tether combinations, including the REEP5–MFN1/2 pair, define the variation in MERCs (Eisenberg-Bord et al., 2016; Iwasawa et al., 2011; Stoica et al., 2014; Szabadkai et al., 2006), our results suggest that mitochondrial ROS production can be altered and regulated by dynamic MERC interactions. This may have a profound impact on the understanding of cell physiology, aging, and a variety of pathological conditions, including neurodegenerative diseases in which ROS changes are frequently observed (Sies et al., 2022; Stefanatos and Sanz, 2018). Our study highlights the importance of organelle membrane contacts in mediating organelle crosstalk and coordination of cellular functions.

## Materials and methods
### Cell culture and transfection
HEK293T cells were obtained from ATCC (CRL-3216), COS-7 cells were a gift from Prof. Liangyi Chen (Peking University, Beijing, China), U2OS cells were from the Cell Bank, Chinese Academy of Sciences (Shanghai, China), MEF cells were a gift from Prof. Li Yu (Tsinghua University, Beijing, China). All cell lines were cultured at 37°C and 5% CO₂ in Dulbecco's Modified Eagle Medium (D6429; Sigma-Aldrich) containing 5% (vol/vol) FBS (SE100-011; Vistech) and 0.2% penicillin–streptomycin (15140122; Gibco). All cell lines were regularly tested for mycoplasma contamination by PCR and/or stained with DAPI. Transient transfections of HEK293T cells using PEI 25K (23966-1;

Polysciences) and other cells were done with Lipofectamine 2000 (11668019; Thermo Fisher Scientific), incubated for 20–24 h prior to the immunoprecipitation or imaging experiments.

All stable cell lines (HaloTag-MFN2, mito-roGFP2, mito-HyPer7, ER-TriPer) were generated by lentivirus transduction. The cDNA sequences were cloned to the lentiviral expression vector (pLVX-IRES-Puro) and delivered into HEK293T cells with packing plasmids (pMD2.G and psPAX2) using PEI 25K. The supernatant containing virus particles was centrifuged at 3,000 rpm for 10 min, filtered, and added into cells at the ratio of 1/3. Cells were selected and maintained by puromycin (A610593; Sangon Biotech) containing media. The lentiviral expression vector and packing plasmids were a kind gift from Prof. Du Feng (Guangzhou Medical University, Guangzhou, China).

For ROS imaging, the cell lines above were incubated with a high glucose medium (D6429; Sigma-Aldrich) containing FBS and penicillin-streptomycin. Except for mito-HyPer7 imaging, media was replaced with HBSS solution (H1025; Solarbio) containing 20 mM HEPES (A600264-0250; Sangon Biotech) 24 h after transfection (Pak et al., 2020) and imaged on LSM 880 microscope (Zeiss) with a 63× oil objective at 37°C.

For assays involving mitochondrial motility, e.g., GI-SIM, photo switch, doxycycline-inducible expression, cells were incubated with low glucose medium (2337699; Gibco) containing 5% FBS (SE100-011; Vistech), 4 mM GlutaMAX (35050061; Gibco), 25 mM HEPES (A600264-0250; Sangon Biotech) pH = 7.4 and 0.2% penicillin-streptomycin (15140122; Gibco) for 3 h before imaging to increase mitochondrial dynamics.

## Reagents and antibodies
Chemical reagents were obtained from the following sources: Rapamycin (HY-10219; MCE), MitoTracker Red CMXRos (M7512; Invitrogen), Janelia Fluor 549 (GA1110; Promega) and 646 (GA1120; Promega) HaloTag Ligands, Doxycycline (S5159; Selleck), and Nocodazole (S1765; Beyotime). Restriction endonucleases were from New England Biolabs and other molecular cloning-related products from Vazyme. The following antibodies were used: rabbit anti-Myc (2278; CST), rabbit anti-Flag (PM020; MBL), rabbit anti-GFP (598; MBL), rabbit anti-MFN1 (14739; CST), rabbit anti-MFN2 (11925; CST), rabbit anti-lgG (30000-0-AP; Proteintech), mouse anti-GAPDH (SC-32233; Santa Cruz), mouse anti-Tom20 (SC-17764; Santa Cruz), rabbit anti-GST (1003-8; HuaBio), rabbit anti-REEP2 (A17152; ABclonal), rabbit anti-REEP3 (1-77105; Novus NBP), rabbit anti-REEP4 (26650-1-AP; Proteintech), rabbit anti-REEP5 (A10392; ABclonal), rabbit anti-REEP6 (FNab07232; FineTest), rabbit anti-Rtn2 (11168-1-AP; Proteintech), rabbit anti-Rtn3 (A302-860A-T; Bethyl Laboratories), and rabbit anti-Rtn4 (FNab07524; FineTest). The secondary antibodies for immunoblotting, IRDye 800CW donkey anti-mouse (926-32212), IRDye 800CW donkey anti-rabbit (926-32213), and IRDye 680CW donkey anti-rabbit (926-68073), were purchased from LI-COR Biosciences. The secondary antibodies for immunostaining, donkey anti-rabbit Alexa Fluor 488 (A32790) and donkey anti-mouse Alexa Fluor 555 (A32773), were purchased from Thermo Fisher Scientific.

## Live-cell imaging
U2OS, COS-7, and MEF cells were seeded on glass-bottomed chambers (C4-1.5H-N; Cellivs). Live-cell imaging was performed on an LSM 880 microscope (Zeiss) with a 63× oil objective at standard confocal resolution or enhanced Airyscan resolution. Live-cell imaging was carried out at 37°C and 5% $CO_2$.

## Immunofluorescence cell staining
Cells were seeded on glass coverslips (WHB-12-CS; WHB), fixed with 4% formaldehyde (P6148; Sigma-Aldrich) for 10 min at room temperature, and incubated with blocking buffer (PBS + 0.4% Triton X-100 + 1% BSA + 0.02% NaN3) for 30 min at room temperature. Fixed cells were then stained with primary antibody at room temperature for 2 h, washed in PBS, and stained with fluorescently labeled secondary antibody for 1 h, washed in PBS, stained with DAPI (C0060; Solarbio), and mounted with Mowiol. The slides were analyzed under the fluorescence microscope (Zeiss).

## GI-SIM imaging
U2OS cells stably expressing HaloTag-MFN2 were transfected with the indicated plasmids and labeled with JF549 ligand following the protocol provided by the manufacturer and then replaced with a low-glucose medium for 3 h before imaging. 500 µM Trolox (10011659; Cayman Chemical) was added to reduce phototoxicity. Cells were maintained at 37°C and 5% $CO_2$ during imaging.

GI-SIM images of U2OS cells were acquired on the Multi-SIM imaging system (NanoInsights) with a 100X/1.49 NA oil objective (Nikon CFI SR HP Apo), solid-state single-mode lasers (488, 561, 640 nm) and a COMS (Complementary Metal-Oxide-Semiconductor) camera (ORCA-Fusion C14440-20UP, Hamamatsu). To obtain optimal images, immersion oils with refractive indices of 1.518 were used for U2OS cells in glass chambers. The microscope was routinely calibrated with 100 nm fluorescent spheres to calculate both the lateral and axial limits of image resolution. SIM image stacks were reconstructed using SI-Recon 2.11.19 (NanoInsights) with the following settings: pixel size 30.6 nm; channel-specific optical transfer functions; Wiener filter constant 0.01 for 2D mode; and discarded negative intensities background. Then the reconstructed SIM image was denoised with total variation (TV) constraint. Pixel registration was corrected to be <1 pixel for all channels using 100 nm fluorescence beads. Multi-SIM images were analyzed by ImageJ (National Institutes of Health).

## Hypotonic cell treatment
COS-7 cells and MEF cells transfected with the indicated plasmids were imaged at 22 h after transfection. During imaging, cells were replaced with hypotonic medium (5% DMEM in water, pH ~7) according to King et al. (2020) and incubated at 37°C with 5% $CO_2$.

## FLIP
All FLIP experiments used a 405 nm laser to repeatedly bleach one subregion within the cell.

For FLIP following hypotonic treatment, transfected MEF cells were captured every 500 ms followed by one round of bleaching in regions away from contact sites. For each round of bleaching, laser power was set at 4% and for 60 iterations.

For FLIP following rapamycin-induced REEP5–MFN1/2 interaction, the same size of subregions of U2OS cells were chosen and captured every 900 ms followed by one round of bleaching. For each round of bleaching, laser power was set at 100% for 200 iterations.

## Photoswitch

MEF cells transfected with mito-Dendra2 were photoswitched by a 405 nm laser. During imaging, the same size of subregions within U2OS cells were randomly chosen and were illuminated at 5% laser power for 120 iterations.

## Co-immunoprecipitation

HEK293T cells were lysed with 150 mM NaCl (10019318; HUSHI), 50 mM Tris-HCl (A600194;BBI) pH = 7.4, and 0.1% NP-40 (A500109; Sangon Biotech), supplemented with protease inhibitor cocktail (C0101; LabLead) on ice. Cells were scraped into Eppendorf tubes and rotated for 40 min at 4°C. The cell lysate was centrifuged at 12,000 rpm for 20 min. For co-immunoprecipitation (Co-IP), the supernatant was transferred to new tubes and incubated with anti-Myc magnetic beads (B26302; Selleck) or anti-Flag affinity gel (B23102; Selleck) and rotated at 4°C overnight. The beads or affinity gel were then washed four times with a buffer (150 mM NaCl, 50 mM Tris-HCl pH = 7.4) and boiled in 2× sample buffer. For endogenous IP, the supernatant was incubated with rabbit anti-MFN1 (14739; CST), rabbit anti-MFN2 (11925; CST), or rabbit anti-IgG (30000-0-AP; Proteintech) at 4°C overnight with gentle agitation, and then incubated with Protein A/G beads (Smart-Lifesciences SA032005) at 4°C for 4 h. The beads were washed twice in the buffer (150 mM NaCl, 50 mM Tris-HCl pH 7.4, 1 mM EDTA, 0.1% NP-40), twice in another buffer (150 mM NaCl, 50 mM Tris-HCl pH 7.4, 1 mM EDTA), and then boiled in 2× sample buffer. Samples were subjected to SDS-PAGE and analyzed by immunoblotting.

## GST-pulldown assay

GST-MFN1/2 C-terminal cytoplasmic domains, His-REEP5, and GST were expressed in BL21(DE3) competent cells. The recombinant proteins were purified by GST resin (J11C; ProbeGene) or Ni-NTA column (SA003100; Smart-Lifesciences). In glutathione-S-transferase (GST) pull-down assays, GST and GST-tagged proteins were applied to GST resin, then incubated with His-tagged proteins in 20 mM Tris-HCl pH 8.0, 50 mM NaCl, 0.1% Triton X-100, 250 mM imidazole (A600277; BBI), 10% glycerol (10010618; Sinopharm) at 4°C for 5 h. After four washes, proteins were eluted and dissolved in sample buffer for SDS-PAGE and analyzed by Coomassie Blue staining or immunoblotting.

## Immunoblotting

Cells were washed with ice-cold PBS and lysed with 150 mM NaCl (10019318; HUSHI), 50 mM Tris-HCl (A600194; BBI) pH = 7.4, 0.1% NP-40 (A500109; Sangon Biotech), supplemented with

protease inhibitor cocktail (C0101; LabLead). Cell lysates were then subjected to SDS-PAGE on 10% or 12% gels and blotted onto the PVDF membrane (IEVH85R; Millipore). Membranes loaded with proteins were blocked with 10% skimmed milk at room temperature for 1 h and incubated with primary antibodies at room temperature for 2 h or overnight at 4°C. After washing, membranes were incubated with secondary antibodies at room temperature for 1 h in the dark, washed, and developed by Odyssey CLx infrared scanner (LI-COR).

## siRNA and transfection

For siRNA-mediated gene knockdown, siRNA duplexes were purchased from GenePharma. REEP5 and Rtn3 were depleted by two sequential knockdowns using Lipofectamine RNAiMAX (13778030; Invitrogen) following the protocol provided by the manufacturer. The working concentration of siRNA was 50 nM for U2OS and 100 nM for HEK293T. Cells were seeded at 40–50% confluency and transfected twice at 24 h intervals. For rescue experiments, plasmids and siRNAs were cotransfected with Lipofectamine 2000 24 h after the first round of siRNA transfection. Cells were analyzed 24 h after the second round of siRNA transfection. Sequences for siRNAs were as follows: si-REEP5#1 5′-GCGUGAACAGGAGCUUCAUTT-3′, si-REEP5#2 5′-GCUGAAUUCUUCUCUGAUATT-3′; si-Rtn3#1 5′-GACCAGCAACAACGUAUCAUT-3′, si-Rtn3#2 5′-GUGGAAGAGCAAAUAGAUATT-3′, si-Rtn3#3 5′-GGUAGAAGGCAUUUAUACATT-3′.

## Quantification of mitochondrial distribution

Quantification of mitochondrial distribution was evaluated by Clock Scan plug-in of ImageJ, as described by Dobretsov et al. (2017). Briefly, confocal images, after applying 31 × 31 median, were followed by Costes' autothreshold subtraction. The boundary of the mitochondrial signal labeled with Tom20 or MitoTracker within the individual cell was outlined using the "polygon" tool as the Region of Interest (ROI). Following the standard "Clock Scan" protocol, X and Y coordinates of the ROI center were set, with "scan limits" at "1," and "real radius" selected to generate the intensity of mitochondrial signal as a function of radial distance to the center of the mitochondrial network. For quantification, 10–17 cells were measured.

## Quantification of ROS levels

U2OS cells stably expressing mito-roGFP2 or mito-HyPer7 were generated and used to quantify the mitochondrial ROS level. For mito-roGFP2 probe, fluorescent images were randomly taken by using 405 and 488 nm lasers for excitation, and 500–530 nm spectral detection, respectively. The laser powers of 405 and 488 nm channels were all set at 0.2%, gain value 650 for every ROS assay. The 405/488 nm excitation ratio of mito-roGFP2 was calculated.

For mito-HyPer7 probe, fluorescent images were randomly taken by using 405 and 488 nm lasers for excitation, and 475–575 nm spectral detection, respectively. The laser powers of 405 and 488 nm channels were all set at 0.3%, gain value of 700 for every ROS assay. The 488/405 nm excitation ratio of mito-HyPer7 was calculated.

U2OS cells stably expressing ER-TriPer were generated and used to quantify ER ROS level. For the ER-TriPer probe, fluorescent images were randomly taken by using 405 and 488 nm lasers for excitation, and 506–568 nm spectral detection, respectively. The laser powers of 405 and 488 nm channels were all set at 6.0% and 4.0%, gain value at 700 for every ROS assay. The 488/405 nm excitation ratio of ER-TriPer was calculated.

## Online supplemental material

Fig. S1 shows additional data for Fig. 1. Fig. S2 shows additional data for Fig. 4. Fig. S3 shows additional data for Fig. 5. Video 1 shows imaging of enriched signals of mEmerald-REEP5 and HaloTag-MFN1/2 at ER tubule-mitochondrial LICV contact sites upon hypotonic treatment. Video 2 shows imaging of enriched signals of mEmerald-REEP5 and HaloTag-MFN1/2 at ER LICV-mitochondrial LICV contact sites upon hypotonic treatment. Video 3 shows imaging of mEmerald-REEP5 signals at contact sites in the absence/presence of MFN2 before and after photobleaching upon hypotonic treatment. Video 4 shows imaging of HaloTag-MFN2 recruiting mEmerald-3XREEP5 C-terminus. Video 5 shows imaging of co-movement events between HaloTag-MFN2 and mEmerald-LAMP1 WT or chimeric mutant. Video 6 shows imaging of ER–mitochondria dynamics in four patterns. Video 7 shows imaging of ER–mitochondrial co-movement in U2OS cells co-expressing REEP5-MFN2 or Sec61β-MFN2. Video 8 shows imaging of mitochondrial directional movement along tubular ER in REEP5-expressing WT MEFs. Video 9 shows imaging of mitochondrial directional movement along tubular ER in REEP5-expressing MFN1&2 DKO MEFs. Video 10 shows imaging of mitochondrial directional movement before and after doxycycline-induced REEP5 expression. Video 11 shows imaging of mitochondrial directional movement before and after doxycycline-induced REEP5 mutant #2 expression. Video 12 shows imaging of mitochondrial directional movement before and after doxycycline-induced REEP6 expression. Video 13 shows imaging of ER-mitochondrial dynamics in U2OS cells with rapamycin-induced REEP5-MFN1 irreversible binding. Video 14 shows imaging of ER–mitochondrial dynamics in U2OS cells with rapamycin-induced REEP5-MFN2 irreversible binding. Video 15 shows imaging of mito-RFP signal intensity dynamics before and after photobleaching in U2OS cells with rapamycin-induced REEP5-MFN1 irreversible binding. Video 16 shows imaging of mito-RFP signal intensity dynamics before and after photobleaching in U2OS cells with rapamycin-induced REEP5-MFN2 irreversible binding.

## Data availability

Original data are available from the corresponding authors upon request.

## Acknowledgments

We thank Prof. Dong Li from the Institute of Biophysics, Chinese Academy of Sciences for his support and suggestions on experimental methods and Multi-SIM imaging techniques, Prof. Li Yu from Tsinghua University for advice and sharing experimental materials, and Prof. Cong Yi, Prof. Qiming Sun, Prof. Yu Feng, and Prof. Hongguang Xia from Zhejiang University for sharing experimental materials and discussions. We thank Yun Feng, Qing Bian, and Xiang Zhang from Center for Biological Imaging (CBI), Institute of Biophysics, Chinese Academy of Sciences for Multi-SIM imaging analysis. We thank Shuangshuang Liu from Imaging Center in Zhejiang University for data analysis and technical support by the Core Facilities in Zhejiang University-University of Edinburgh (ZJU-UoE) Institute.

This study was supported by grants from the Natural Science Foundation of China (NSFC; 32070700). Open Access funding provided by University of Edinburgh.

Author contributions: S. Chen: Conceptualization, Data curation, Formal analysis, Investigation, Methodology, Project administration, Validation, Visualization, Writing—original draft, Writing—review & editing, Y. Sun: Formal analysis, Investigation, Validation, Visualization, Y. Qin: Investigation, L. Yang: Investigation, Validation, Visualization, Z. Hao: Investigation, Visualization, Z. Xu: Investigation, M. Bjorklund: Writing—review & editing, W. Liu: Supervision, Writing—review & editing, Z. Hong: Conceptualization, Data curation, Formal analysis, Funding acquisition, Investigation, Methodology, Project administration, Resources, Supervision, Validation, Visualization, Writing—original draft, Writing—review & editing.

Disclosures: The authors declare no competing interests exist.

Submitted: 8 April 2023

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

# Supplemental material

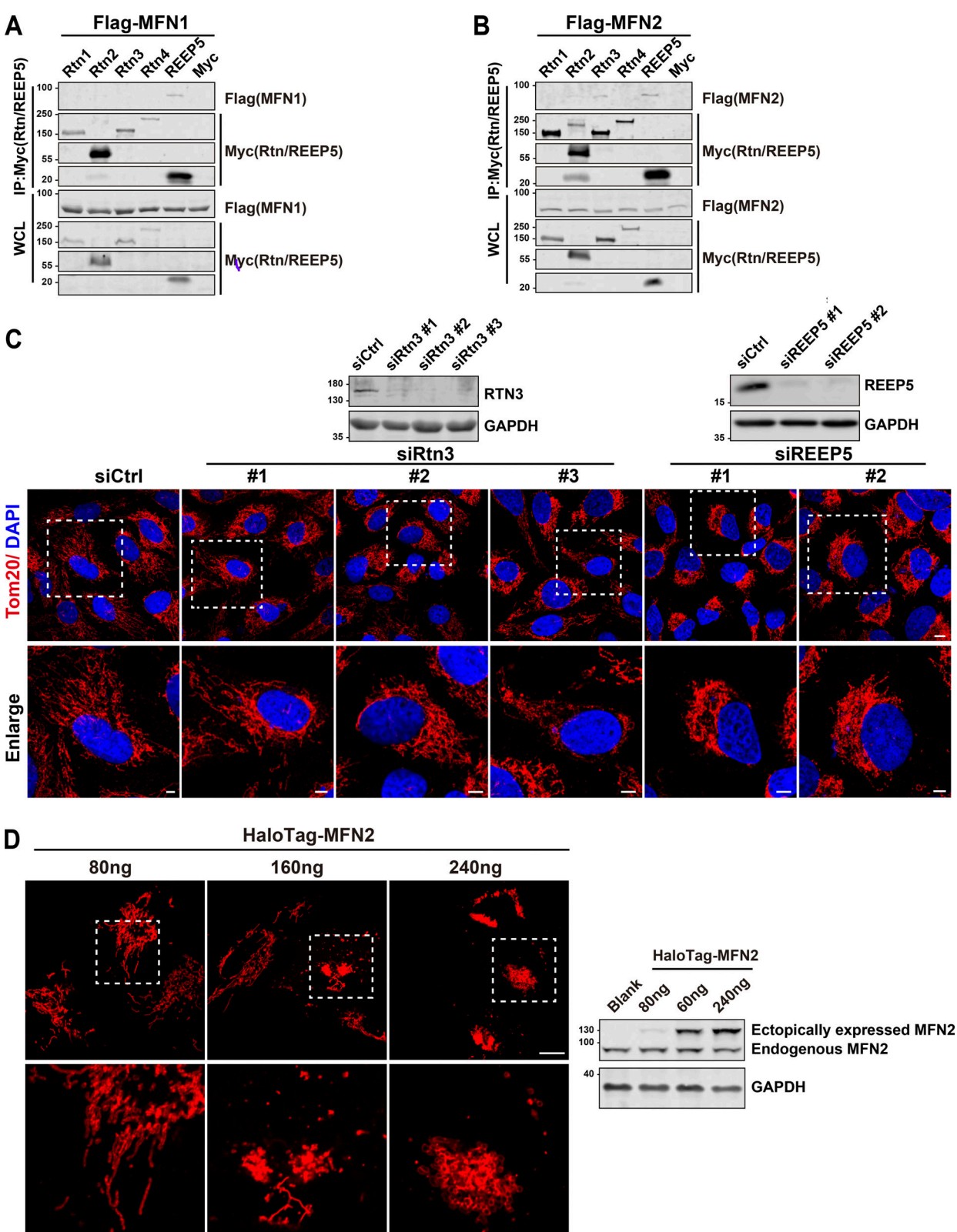

Figure S1. **REEP5 specifically interacts with MFN1/2, and the REEP5 knockdown affects mitochondrial morphology. (A and B)** Co-IP analysis of Myc-tagged Rtn1-4 or REEP5 with Flag-MFN1/2 from HEK293T cells. **(C)** Confocal images of mitochondrial distribution in U2OS cells transfected with siRtn3 or siREEP5. siRNA efficiencies were analyzed by immunoblotting. Endogenous mitochondria were immunostained with anti-Tom20 and the nucleus was labeled with DAPI. Bar: 10 μm (whole cell view); 5 μm (zoomed-in images). **(D)** Mitochondrial morphology from U2OS cells expressing different amounts of HaloTag-MFN2 as indicated. MFN2 protein levels were detected by immunoblotting. Cell number for each transfection, $4 \times 10^5$. Bar: 10 μm. Source data are available for this figure: SourceData FS1.

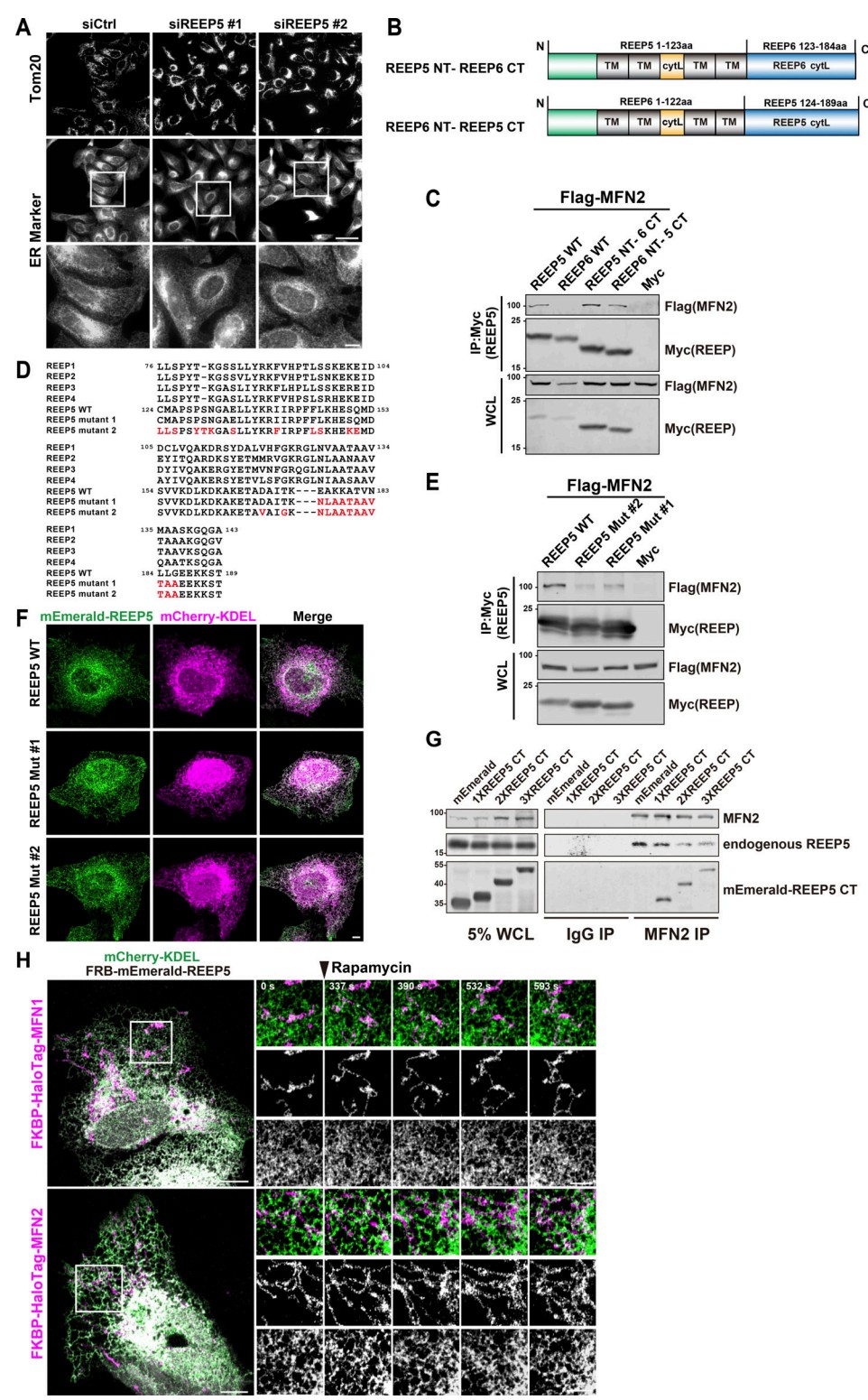

Figure S2.   **Mitochondrial morphology is disrupted by reduced or enhanced REEP5–MFN1/2 interactions. (A)** ER morphology in REEP5 depleted cells. Endogenous mitochondria and ER were immunostained in U2OS cells. Bars: 20 µm (whole cell view); 5 µm (zoomed-in images). **(B)** Schematic diagram of REEP chimeric mutants. Residue numbers are shown as indicated. NT, N-terminus; CT, C-terminus; TM, transmembrane; cytL, cytosolic loop. **(C)** Co-IP analysis of Myc-tagged REEP chimera with Flag-MFN2 from HEK293T cells. **(D)** Alignment of sequences from REEP1-5 and REEP5 mutants. Positions of the mutations were indicated in red and residue numbers were shown as indicated. **(E)** Co-IP analysis of Myc-tagged REEP5 mutants with Flag-MFN2 from HEK293T cells. **(F)** Confocal images of mEmerald-REEP5 mutants and mCherry-KDEL expressing U2OS cells. Bar: 5 µm. **(G)** Co-IP analysis of endogenous MFN2 with REEP5 from HEK293T cells expressing mEmerald tagged REEP5 CT. **(H)** Time-lapse images of ER–mitochondria dynamics in U2OS cells with rapamycin-induced REEP5–MFN1/2 binding (also see Videos 13 and 14). 20 nM rapamycin was added at the time point indicated. Bars: 10 µm (whole cell view); 5 µm (zoomed-in images). Source data are available for this figure: SourceData FS2.

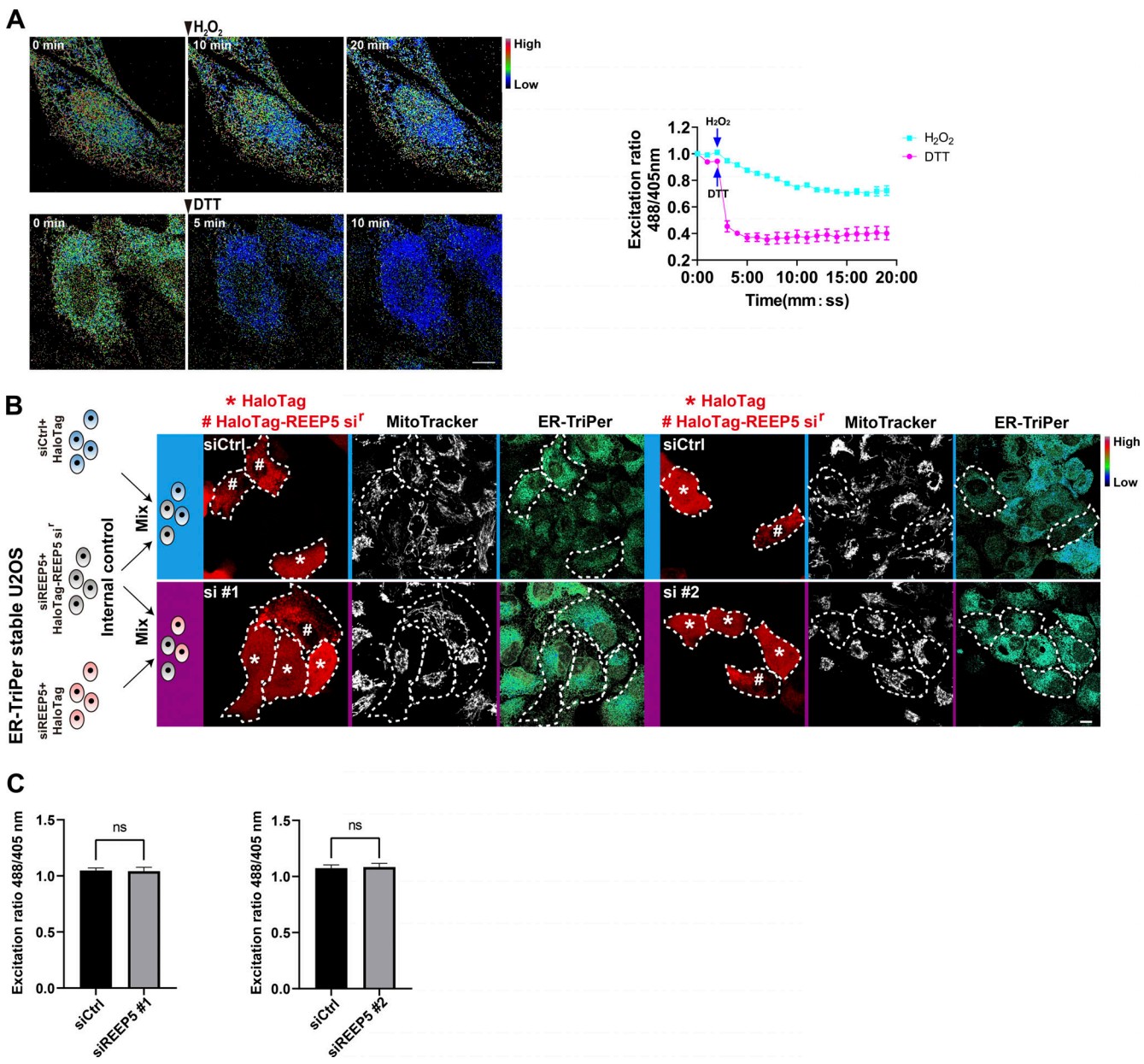

Figure S3. **Increased ROS production in REEP5 knockdown cells is independent of ER origin. (A)** Time course of TriPer fluorescence excitation (488/405 nm) ratiometric images (R488/405) of ER-TriPer stable U2OS cells after addition of 200 µM $H_2O_2$ or 2 mM DTT for the indicated time. The ratiometric changes overtime were calculated and shown in the graph. Bar: 10 µm. **(B)** Schematic workflow and images of ER $H_2O_2$ in REEP5 depleted U2OS cells. Top panel, siCtrl + internal control; bottom panel, siREEP5 #1 or #2 + internal control. Bars: 10 µm. **(C)** Graph of excitation ratio (R488/405 nm) quantification from B. Values shown are means ± SEM, $n$ = 7 for A; $n$ = 20 for C.

Video 1. **Imaging of enriched signals of mEmerald-REEP5 and HaloTag-MFN1/2 at ER tubule-mitochondrial LICV contact sites upon hypotonic treatment, corresponding to** Fig. 2 A**.** Time-lapse images showing over 3 min of two live COS-7 cells transfected with HaloTag-MFN1/2 (red) and mEmerald-REEP5 (green). Hypotonic solutions were added during imaging. White arrowheads indicate the enrichment of REEP5 and MFN1/2 signals at contact sites. Frames were captured every 2 s or 3 s using a laser-scanning confocal microscope. The video is shown at 5 frames/s. Bar: 1 µm.

**Video 2. Imaging of enriched signals of mEmerald-REEP5 and HaloTag-MFN1/2 at ER LICV-mitochondrial LICV contact sites upon hypotonic treatment, corresponding to** Fig. 2 A**.** Time-lapse images showing over 3 min of two live COS-7 cells transfected with HaloTag-MFN1/2 (red) and mEmerald-REEP5 (green). Hypotonic solutions were added during imaging. White arrowheads indicate the enrichment of REEP5 and MFN1/2 signals at contact sites. Frames were captured every 3 s using a laser-scanning confocal microscope. The video is shown at 5 frames/s. Bar: 1 μm.

**Video 3. Imaging of mEmerald-REEP5 signals at contact sites in the absence/presence of MFN2 before and after photobleaching upon hypotonic treatment, corresponding to** Fig. 2 B**.** Time-lapse images showing 5 s of four live MFN1&2 DKO MEF cells transfected with mEmerald-REEP5/sec61β (green), mito-RFP (red), and HaloTag-MFN2/HaloTag. Imaging started at 10 min after hypotonic treatment. Dash-line boxes indicate bleached regions. White arrowheads indicate contact sites. Frames were captured every 0.5 s using a laser-scanning confocal microscope. The video is shown at 3 frames/s. Bars: 1 μm.

**Video 4. Imaging of HaloTag-MFN2 recruiting mEmerald-3XREEP5 C-terminus, corresponding to** Fig. 2 C**.** Time-lapse images showing 42 s of a live MFN1&2 DKO MEF cell transfected with mEmerald-3XREEP5 C-terminus (green), HaloTag-MFN2 (red), and mCherry-KDEL (grey). White arrowheads indicate that 3XREEP5 C-terminus was recruited onto MFN2-labeled mitochondria but not ER. Frames were captured every 3 s using a laser-scanning confocal microscope. The video is shown at 2 frames/s. Bars: 5 μm for whole cell view; 1 μm (zoomed-in images).

**Video 5. Imaging of co-movement events between HaloTag-MFN2 and mEmerald-LAMP1 WT or chimeric mutant, corresponding to** Fig. 2 D**.** Time-lapse images showing 48 s of two live U2OS cells transfected with mEmerald-LAMP1 WT/mutant (green) and HaloTag-MFN2 (red). White arrowheads indicate co-movement events. Frames were captured every 2 s using a laser-scanning confocal microscope. The video is shown at 2 frames/s. Bars: 5 μm.

**Video 6. Imaging of ER-mitochondria dynamics in four patterns, corresponding to** Fig. 3 A**.** Time-lapse images showing 7 min 30 s of live U2OS cells transfected with mEmerald-REEP5 (green) and HaloTag-MFN2 (red). White arrowheads indicate contact sites, yellow arrowheads indicate ER-mitochondria dissociation sites. Frames were captured every 10 s using a multimodality structured illumination super-resolution microscopy. The video is shown at 2 frames/s. Bars: 0.5 μm.

**Video 7. Imaging of ER–mitochondrial co-movement in U2OS cells co-expressing REEP5-MFN2 or Sec61β-MFN2, corresponding to** Fig. 3 B**.** Time-lapse images showing 13 min 20 s of two live U2OS cells transfected with mEmerald-REEP5 (green) and HaloTag-MFN2 (red). White arrowheads indicate ER–mitochondrial co-movements. Frames were captured every 10 s using a multimodality structured illumination super-resolution microscopy. The video is shown at 5 frames/s. Bars: 5 μm.

**Video 8. Imaging of mitochondrial directional movement along tubular ER in REEP5-expressing WT MEFs, corresponding to** Fig. 3 F**.** Time-lapse images showing 4 min 30 s of two live WT MEF cells transfected with mEmerald-REEP5/mEmerald (cyan), mito-RFP (red), and HaloTag-KDEL (green). Frames were captured every 9 s using a laser-scanning confocal microscope. The video is shown at 7.5 frames/s. Bars: 2 μm.

**Video 9. Imaging of mitochondrial directional movement along tubular ER in REEP5-expressing MFN1&2 DKO MEFs, corresponding to** Fig. 3 G**.** Time-lapse images showing 4 min 30 s of two live MFN1&2 DKO MEF cells transfected with mEmerald-REEP5/mEmerald (cyan), mito-RFP (red), and HaloTag-KDEL (green). Frames were captured every 9 s using a laser-scanning confocal microscope. The video is shown at 7.5 frames/s. Bars: 2 μm.

**Video 10. Imaging of mitochondrial directional movement before and after doxycycline-induced REEP5 expression, corresponding to** Fig. 4 M**.** Time-lapse images showing over 9 min of a live U2OS cell transfected with TetOn3G-mEmerald-REEP5, HaloTag-MFN2 (red), and BFP-KDEL. 200 nM doxycycline and 5 uM Trolox were added before imaging. Cells were fast imaged for 10 min before REEP5 expression (left) and 10 min after REEP5 expression (right). White arrowheads indicate mitochondrial directional movements. Frames were captured every 15 s using a laser-scanning confocal microscope. The video is shown at 3 frames/s. Bars: 5 μm.

Video 11.   **Imaging of mitochondrial directional movement before and after doxycycline-induced REEP5 mutant #2 expression, corresponding to Fig. 4 M.** Time-lapse images showing over 9 min of a live U2OS cell transfected with TetOn3G-mEmerald-REEP5 mut #2, HaloTag-MFN2 (red) and BFP-KDEL. 200 nM doxycycline and 5 µM Trolox were added before imaging. Cells were fast imaged for 10 min before REEP5 mut #2 expression (left) and 10 min after REEP5 mut #2 expression (right). White arrowheads indicate mitochondrial directional movements. Frames were captured every 15 s using a laser-scanning confocal microscope. The video is shown at 3 frames/s. Bars: 5 µm.

Video 12.   **Imaging of mitochondrial directional movement before and after doxycycline-induced REEP6 expression, corresponding to Fig. 4 M.** Time-lapse images showing over 9 min of a live U2OS cell transfected with TetOn3G-mEmerald-REEP6, HaloTag-MFN2 (red) and BFP-KDEL. 200 nM doxycycline and 5 µM Trolox were added before imaging. Cells were fast imaged for 10 min before REEP6 expression and 10 min after REEP6 expression. White arrowheads indicate mitochondrial directional movements. Frames were captured every 15 s using a laser-scanning confocal microscope. The video is shown at 3 frames/s. Bars: 5 µm.

Video 13.   **Imaging of ER-mitochondrial dynamics in U2OS cells with rapamycin-induced REEP5-MFN1 irreversible binding, corresponding to Fig. S2 H.** Time-lapse images showing over 12 min of a live U2OS cell transfected with FRB-mEmerald-REEP5, FKBP-HaloTag-MFN1 (red), and mCherry-KDEL (green). 20 nM rapamycin was added at the indicated time point. Frames were captured every 19.2 s using a laser-scanning confocal microscope. The video is shown at 80 frames/s. Bars: 2 µm.

Video 14.   **Imaging of ER-mitochondrial dynamics in U2OS cells with rapamycin-induced REEP5-MFN2 irreversible binding, corresponding to Fig. S2 H.** Time-lapse images showing over 12 min of a live U2OS cell transfected with FRB-mEmerald-REEP5, FKBP-HaloTag-MFN2 (red), and mCherry-KDEL (green). 20 nM rapamycin was added at the indicated time point. Frames were captured every 19.2 s using a laser-scanning confocal microscope. The video is shown at 80 frames/s. Bars: 2 µm.

Video 15.   **Imaging of mito-RFP signal intensity dynamics before and after photobleaching in U2OS cells with rapamycin-induced REEP5-MFN1 irreversible binding, corresponding to Fig. 4 N.** Time-lapse images showing over 15 s of two live U2OS cells transfected with FRB-mEmerald-REEP5, FKBP-HaloTag-MFN1, and mito-RFP (gray). 20 nM rapamycin was added to induce REEP5-MFN1 irreversible binding and DMSO was used as negative control. Dash-line boxes indicate bleached regions. Frames were captured every 0.9 s using a laser-scanning confocal microscope. The video is shown at 5 frames/s. Bars: 5 µm.

Video 16.   **Imaging of mito-RFP signal intensity dynamics before and after photobleaching in U2OS cells with rapamycin-induced REEP5-MFN2 irreversible binding, corresponding to Fig. 4 O.** Time-lapse images showing over 15 s of two live U2OS cells transfected with FRB-mEmerald-REEP5, FKBP-HaloTag-MFN2, and mito-RFP (gray). 20 nM rapamycin was added to induce REEP5-MFN2 irreversible binding and DMSO was used as a negative control. Dash-line boxes indicate bleached regions. Frames were captured every 0.9 s using a laser-scanning confocal microscope. The video is shown at 5 frames/s. Bars: 5 µm.

