## [Peer Review File · The Journal of Cell Biology]

Dynamic interaction of REEP5-MFN1/2 enables mitochondrial hitchhiking on tubular ER

Shue Chen, Yang Sun, Yuling Qin, Lan Yang, Zhenhua Hao, Zhihao Xu, Mikael Bjorklund, Wei Liu, and Zhi Hong

Corresponding Author(s): Zhi Hong, Second Affiliated Hospital of Zhejiang University

Review Timeline:

Submission Date:	2023-04-08
Editorial Decision:	2023-05-15
Revision Received:	2024-02-15
Editorial Decision:	2024-04-12
Revision Received:	2024-05-28

Monitoring Editor: Laura Lackner

Scientific Editor: Dan Simon

Transaction Report:

DOI: <https://doi.org/10.1083/jcb.202304031>

May 15, 2023

Re: JCB manuscript #202304031

Dr. Zhi Hong
Zhejiang University
718 East Haizhou Rd., Haining, Zhejiang, P.R. China
Haining 314400
China

Dear Dr. Hong,

Thank you for submitting your manuscript entitled "Dynamic interaction of REEP5-MFN1/2 enables mitochondrial hitchhiking on tubular ER." Your manuscript has been assessed by expert reviewers, whose comments are appended below. Although the reviewers express potential interest in this work, significant concerns unfortunately preclude publication of the current version of the manuscript in JCB.

You will see that the reviewers feel that your study would potentially be important as it reveals a novel ER-mitochondria tethering complex. However, they also raise several important concerns that will all need to be addressed in full and will require additional experiments. In particular it would be essential to add new data that definitively shows that REEP5 on ER and MFN1/2 on mitochondria form tethering complexes and also to investigate in more detail the role of this interaction in the maintenance of mitochondrial ROS levels.

Please let us know if you are able to address the major issues outlined above and wish to submit a revised manuscript to JCB. Note that a substantial amount of additional experimental data likely would be needed to satisfactorily address the concerns of the reviewers.

The typical timeframe for revisions is three to four months. While most universities and institutes have reopened labs and allowed researchers to begin working at nearly pre-pandemic levels, we at JCB realize that the lingering effects of the COVID-19 pandemic may still be impacting some aspects of your work, including the acquisition of equipment and reagents. Therefore, if you anticipate any difficulties in meeting this aforementioned revision time limit, please contact us and we can work with you to find an appropriate time frame for resubmission. Please note that papers are generally considered through only one revision cycle, so any revised manuscript will likely be either accepted or rejected.

If you choose to revise and resubmit your manuscript, please also attend to the following editorial points. Please direct any editorial questions to the journal office.

GENERAL GUIDELINES:

Text limits: Character count for a Report is < 20,000; a full Research Article is < 40,000, not including spaces. Count includes title page, abstract, introduction, the joint Results & Discussion, and acknowledgments. Count does not include materials and methods, figure legends, references, tables, or supplemental legends.

Figures: A Report may include up to 5 main text figures; a full Research Article may have up to 10 main text figures. To avoid delays in production, figures must be prepared according to the policies outlined in our Instructions to Authors, under Data Presentation, <https://jcb.rupress.org/site/misc/ifora.xhtml>. All figures in accepted manuscripts will be screened prior to publication.

*****IMPORTANT:** It is JCB policy that if requested, original data images must be made available. Failure to provide original images upon request will result in unavoidable delays in publication. Please ensure that you have access to all original microscopy and blot data images before submitting your revision. ***

Supplemental information: There are strict limits on the allowable amount of supplemental data. Reports may have up to 3 supplemental figures; a full Research Article may have up to 5 supplemental figures. Up to 10 supplemental videos or flash animations are allowed. A summary of all supplemental material should appear at the end of the Materials and methods section.

Please note that JCB now requires authors to submit Source Data used to generate figures containing gels and Western blots with all revised manuscripts. This Source Data consists of fully uncropped and unprocessed images for each gel/blot displayed in the main and supplemental figures. Since your paper includes cropped gel and/or blot images, please be sure to provide one Source Data file for each figure that contains gels and/or blots along with your revised manuscript files. File names for Source Data figures should be alphanumeric without any spaces or special characters (i.e., SourceDataF#, where F# refers to the associated main figure number or SourceDataFS# for those associated with Supplementary figures). The lanes of the gels/blots

should be labeled as they are in the associated figure, the place where cropping was applied should be marked (with a box), and molecular weight/size standards should be labeled wherever possible. Source Data files will be made available to reviewers during evaluation of revised manuscripts and, if your paper is eventually published in JCB, the files will be directly linked to specific figures in the published article.

If you choose to resubmit, please include a cover letter addressing the reviewers' comments point by point. Please also highlight all changes in the text of the manuscript.

Regardless of how you choose to proceed, we hope that the comments below will prove constructive as your work progresses. We would be happy to discuss them further once you've had a chance to consider the points raised. You can contact the journal office with any questions, cellbio@rockefeller.edu or call (212) 327-8588.

Thank you for thinking of JCB as an appropriate place to publish your work.

Sincerely,

Laura Lackner, PhD
Monitoring Editor
Journal of Cell Biology

Dan Simon, PhD
Scientific Editor
Journal of Cell Biology

Reviewer #1 (Comments to the Authors (Required)):

This paper by Chen et al discovers a potentially novel ER-mitochondrial tether that mediates the distribution of mitochondria in mammalian cells. This tethering is proposed to be mediated by a direct interaction between ER localized REEP5 and mitochondrial Mfn1/2. The data that support this are: 1) a direct binding interaction between Mfn1/2 and REEP5 but not the other REEPs; 2) a FLIP experiment showing that colocalized REEP5 and Mfn1/2 are less mobile than other markers; 3) REEP5 knockdown causing perinuclear clustering of mitochondria; and 4) overexpression of REEP5 increasing directional movement of mitochondria in wt but not Mfn null cells. The authors conclude that a REEP5-Mfn1/2 interaction mediates the distribution of mitochondria through a "hitchhiking" mechanism. They then show that the REEP5-Mfn1/2 interaction has to be dynamic, as forced dimerization of REEP5 and Mfn1/2 led to hyper fusion of mitochondria. Finally, silencing REEP5 increased mitochondrial ROS production and forced dimerization reduced ROS, suggesting that the newly identified tether minimizes ROS.

Major points:

This is an intriguing study that may reveal a new ER-mito tethering complex. The experiments seem to be of high quality overall, especially given the more challenging imaging assays. The most compelling aspect of the study are the interaction assays in figure 1. These are convincing, as the interaction seems quite specific, with both Mfn1 and Mfn2 interacting with REEP5, but not REEPs 1, 2 or 6. Also, the mapping of the interaction to the C-termini of both REEP5 and Mfn1/2 are compelling. However, while the remaining experimental results are consistent with the claim that REEP5 and Mfn1/2 function as a tether, no one experiment convincingly demonstrates direct organelle tethering by these two proteins. If the tether model can be more strongly supported, the paper would provide a significant advance appropriate for JCB.

Major points:

1. In figure 3c showing altered mito distribution caused by REEP5 knockdown, co-staining of an independent endogenous ER marker is required to show that the effect of the REEP5 knockdown is specifically on mito distribution, rather than on ER morphology more generally. The latter could alter mito distribution regardless of whether tethering is mediated by REEP5. Also, rescue of mito distribution with wild type but not a C-terminal truncation of REEP5 would make the tether model much more convincing. The authors use siRNA and rescue in a later experiment (figure 4d), so this seems feasible.

2. The main conclusion of the paper is that REEP5 and Mfn1/2 form a tethering complex for ER and mito. While the FLIP experiment in figure 1H is suggestive, it is one example in one cell, and a convincing demonstration that the two proteins form a tether is lacking. If the C-terminus of REEP5 tethers mito to ER through binding to Mfn1/2, then putting the REEP5 C-terminus onto some other protein localized to a different organelle would be predicted to tether mito to that other organelle. Alternatively, overexpressing a cytoplasmic version of the REEP5 C-terminus should disrupt endogenous REEP5-Mfn1/2 tethers and cause loss of directional movement and the collapse of mito as measured in figures 2a,b and 3c. Moreover, given what appears to be a

robust interaction between REEP5 and Mfn1/2, one would expect the free C-terminal fragment to localize to mitochondria through interaction with Mfn1/2.

Other points:

1. In figure 3d,e, one worries that the number of cases of directional movement in the 5min window and in so few cells (n=4) analyzed is so low, hovering between 1 and 2. It seems that a longer window revealing more events would provide higher confidence.
2. In figure 3f,g, it is unclear why expression of REEP6 causes a decrease in directional movement.
3. In figure 4c,d, the cells marked as expressing *Halo or #Halo-REEP5 are not expressing nearly as much protein as the cells to which they are being compared to.

Minor points:

1. In figure 1d, the cartoon of Mfn1/2 risks giving the wrong impression that the proteins have a similar structure or architecture to REEP5. Given that Mfn structure is known, the cartoon should more accurately reflect its domain structure.

Reviewer #2 (Comments to the Authors (Required)):

The ER-mito contact plays important physiological roles. In this manuscript, the authors propose that ER tubule-forming protein REEP5 forms physical interactions with OMM fusogen MFNs. The interactions were thought to promote mitochondrial movement along ER tubules, in conjunction with microtubule. In addition, the link was expected to have a role in mitochondrial ROS regulation. Given that the actual tether between ER and mito in higher eukaryotes is highly debatable, the findings, if proven true, would expand our knowledge on ER-mito communication and coordination. However, the reported observation was largely dependent on overexpressed proteins or artificial conditions. Critical and suitable controls are missing in several key experiments, raising concerns about the fidelity and relevance of the proposed mechanism.

1. Co-IP between REEP/RTN and MFN were done mostly with overexpression. Notably, endogenous antibodies for both MFNs and several REEP/RTNs are commercially available. In endogenous co-IP shown in Fig. 1F, the setup is overly simplified. Whether MFN antibodies/beads have non-specific interactions with REEP5 is not controlled. Co-IP of other REEPs or RTNs was not compared.
2. In Fig. S1A and B, the authors showed that RTN3 had equivalent interactions with MFNs when compared to that of REEP5. However, when RTN3 was deleted in U2OS cells, the mitochondrial defects were not even close to that of REEP5. Could the authors comment on this discrepancy? Along this line, the KD efficiency in these cases were verified at neither protein nor transcript levels, and the depletion specificity was not verified by rescue experiments.
3. The selectivity of REEP5 by MFNs is striking and surprising, considering the sequence similarity between REEPs, or at least with REEP6. By truncation experiments, the authors showed that the C-terminal region of REEP5 is involved in direct interactions. It would thus make sense to construct a chimeric REEP with the tail of REEP5 replaced by that of REEP6. Such a chimera would be expected to maintain tubule-forming ability but lose mitochondrial contact. If so, the chimera could serve as a much better control in several tests, including mitochondrial positioning/movement and ROS regulation.
4. The experiments shown in Fig. 1H is confusing and likely over-interpreted. FLIP experiments are usually informative in testing membrane continuity. In this case, the setup sounds more like a FRAP assay. Importantly, previous FRAP assays have shown that tubule-forming proteins are slower in mobility when compared to that of Sec61b. Therefore, the results could simply be read as their general mobility, but not their preference for contact. If the authors insist on this assay, they should compare the FRAP signal of REEP5 at the contact site in the presence or absence of MFNs, and Sec61b can now serve as a control for not sensitive to such changes. Finally, if the authors really meant for FRAP, the data should be analyzed as curves, but not representative images.
5. Finally, several test conditions are considered very harsh to cells. For example, the overexpression of artificial ER-mito tether itself induces abnormality in mitochondrial morphology. The 1 mM H₂O₂ treatment should certainly cause a drastic fragmentation of mitochondria. Cautions should be taken when performing these assays or interpreting these results.

Reviewer #3 (Comments to the Authors (Required)):

Chen et al. investigate a potential novel interaction of mitofusins with an ER protein, REEP5. This is based on the exclusive localization of transfected wild type, full-length Mfn2 to mitochondria. The authors hypothesize that REEP or reticulon proteins interact with mitochondrial Mfns. They test this by co-immunoprecipitation. REEP5 was found as an interactor for both Mfn1 and Mfn2. Rtn3 interacts at a lower level with Mfn2. The REEP5 protein knockdown accordingly led to altered mitochondrial structure.

Co-localization and interaction were further demonstrated by microscopic techniques using over-expressed fluorescent proteins. Over-expression of REEP5 increased co-migration of mitochondria along the ER as well as diffusion within the mitochondrial network.

The manuscript identifies a novel interaction between Mfn2 and ER network proteins. The majority of the paper uses microscopy and the strength of the interaction remains unclear. Also, under which condition do these proteins interact with each other?

Specific Points:

1. It remains unclear whether Mfn2 (or some of the many fusion constructs used in the study) localizes also to the ER under the conditions used in the manuscript. This was recently demonstrated in PMID: 34296790. It is therefore unclear where the described interaction takes place. Is it on the ER or on mitochondria?
2. The authors claim that the Mfn2-REEP5 interaction regulates mitochondrial ROS levels. However, they cannot tell from their findings whether ROS were made within the mitochondria or elsewhere, e.g., the ER.
3. The sensitivity of the ROS probe used is insufficient. The HyPer7 is the gold standard for measuring mitochondrial ROS with high levels of accuracy and would be more suitable for the experiments performed here. More physiological treatments should be used to provide some information about the level of ROS increases or decreases described in the manuscript. Mfn2 KO or knockdown should also be measured in terms of ROS.
4. Can the interaction be disrupted by expressing fragments of the postulated interaction domains?

Minor Points:

1. The co-immunoprecipitation of endogenous proteins is of unclear significance, since the relative input is not defined.
2. The mitochondria-ER contacts are defined with an unusual abbreviation. Most commonly, these are referred to as MERCs (or biochemically as MAMs).

Reviewer #1

Major points:

“1. In figure 3c showing altered mito distribution caused by REEP5 knockdown, co-staining of an independent endogenous ER marker is required to show that the effect of the REEP5 knockdown is specifically on mito distribution, rather than on ER morphology more generally. The latter could alter mito distribution regardless of whether tethering is mediated by REEP5. Also, rescue of mito distribution with wild type but not a C-terminal truncation of REEP5 would make the tether model much more convincing. The authors use siRNA and rescue in a later experiment (figure 4d), so this seems feasible.”

We agree with the comment that “ER morphology could alter mitochondrial distribution regardless of whether tethering is mediated by REEP5”. To address this issue, we analyzed mitochondrial distribution and ER morphology concurrently in WT and REEP5 knockdown U2OS cells. As shown in the revised Fig. S2A, no obvious changes in global ER morphology were observed, indicating that REEP5 depletion *per se* does not disrupt ER status. Although it has been reported that REEP5 depletion leads to sarco-endoplasmic reticulum vacuolization and cardiac defects (PMID: 32075961) in a mouse model, ER morphology and function in other tissues appear normal, suggesting a potential second-order effect in cardiomyocyte or cell type variations.

To test the specificity of the REEP5-MFN interaction, we derived MFNs-binding-deficient REEP5 mutants. Among the 6 REEP family members, only REEP5 exhibited specific interaction with MFNs via its C-terminus. We generated two chimeric constructs: 1. REEP5 N-terminus + REEP6 C-terminus and 2. REEP6 N-terminus + REEP5 C-terminus (revised Fig. S2B). Surprisingly, both chimeras are able to interact with MFN2 (revised Fig. S2C). One possibility is that the N-terminal portion of REEP6 can potentially mask its C-terminal domain function, preventing its interaction with MFN1/2.

To further examine the selectivity of REEP5 by MFN1/2, we mutated REEP5 into a more REEP1-4-like sequence (revised Fig.S2D). We designed several point mutations intended to disrupt cAPH and CTH, the two domains in the C-terminus of REEP5 (revised Fig. S2D). Mutations in the CTH domain (REEP5 mut#1) affected the interaction substantially, whereas additional mutations in the cAPH domain (REEP5 mut#2) further abolished its interaction with MFN2 (Fig. S2E). Importantly, both mutants display ER patterns consistent with unperturbed wild-type cells, suggesting that they still maintain the ER-tubule forming ability (Fig. S2F).

To conduct the rescue experiment suggested by the reviewer, wild-type REEP5 and the interaction-defective mutant 2 were expressed in REEP5-depleted cells. Using an amino acids substitution mutant rather than the REEP5 C-terminal truncation affords a more parallel comparison between wild-type REEPS and its MFN1/2 interaction-defective derivative. Result of the rescue experiment showed that the mitochondrial phenotype is

markedly reversed by wild-type REEP5, but not the non-binding REEP5 mutant 2 (revised Fig. 4E and F).

Collectively, these additional data further demonstrate that the aberrant mitochondrial distribution we have observed is derived from the loss of the REEP5-MFN2 interaction, rather than indirectly from disruption of ER morphology.

“2. The main conclusion of the paper is that REEP5 and Mfn1/2 form a tethering complex for ER and mito. While the FLIP experiment in figure 1H is suggestive, it is one example in one cell, and a convincing demonstration that the two proteins form a tether is lacking. If the C-terminus of REEP5 tethers mito to ER through binding to Mfn1/2, then putting the REEP5 C-terminus onto some other protein localized to a different organelle would be predicted to tether mito to that other organelle. Alternatively, overexpressing a cytoplasmic version of the REEP5 C-terminus should disrupt endogenous REEP5-Mfn1/2 tethers and cause loss of directional movement and the collapse of mito as measured in figures 2a,b and 3c.

Moreover, given what appears to be a robust interaction between REEP5 and Mfn1/2, one would expect the free C-terminal fragment to localize to mitochondria through interaction with Mfn1/2.”

We appreciate these insightful suggestions and performed three experiments accordingly.

a. To test whether the C-terminus of REEP5 is sufficient for its MFN1/2 interaction in cells, MFN1/2 double knockout (DKO) MEFs were transfected to express monomeric or 2, 3 tandem copies of REEP5 C-terminus (revised Fig. 2C), while membrane association of the monomeric protein is detectable but moderate in the presence of MFN2, 2 and 3 tandem copies of the C-terminus exhibited significant recruitment onto mitochondria when MFN2 is present (revised Fig. 2C and new Video 4). Importantly, these structures are negative for ER marker- KDEL (new Video 4), reflective of mitochondria-recruited REEP5 C-terminus protein.

b. To further validate the tethering function of the REEP5 C-terminus, we replaced the cytoplasmic portion of LAMP1 with 3 copies of the REEP5 C-terminus. Expression of the chimera but not wildtype LAMP1 together with MFN2 indeed increased the contact frequency of mitochondria with the lysosome (revised Fig. 2D and E, and new Video 5), consistent with the idea that REEP5 and MFN2 form a tethering complex.

c. We overexpressed REEP5 C-terminus in 293T cells to compete the endogenous REEP5-MFN2 interaction (revised Fig. S2G). Ectopically expressed REEP5 C-terminus in U2OS cells caused the collapse of mitochondria into peri-nuclear regions (revised Fig. 4G), again suggesting that the REEP5 C-terminus is sufficient in mitochondria association.

We believe that these new data further strengthened the specificity and functionality of the MFN2-REEP5 interaction in ER-mitochondria tethering.

Other points:

“1. In figure 3d,e, one worries that the number of cases of directional movement in the 5min window and in so few cells (n=4) analyzed is so low, hovering between 1 and 2. It seems that a longer window revealing more events would provide higher confidence. ”

Following this suggestion, we performed additional experiments (revised Fig. 4I-L) by increasing the monitored cell numbers from 4 to 10 and expanding the video capturing duration from 5 to 10 min. In addition, we included MFN2-binding deficient REEP5 mutant in this assay as a negative control. These new imaging data improved the quantitative power of this experiment.

“2. In figure 3f,g, it is unclear why expression of REEP6 causes a decrease in directional movement.”

We also noted the same decrease in directional movement from REEP6 expression. However, the REEP6 effect is relatively modest compared to that of the REEP5 expression. We speculate that REEP6 overexpression might interfere with the endogenous REEP5 distribution, or affect the REEP5 protein level at the MERCs.

*“3. In figure 4c,d, the cells marked as expressing *Halo or #Halo-REEP5 are not expressing nearly as much protein as the cells to which they are being compared to. ”*

We performed additional analyses and found more comparable cell images. In the revision, we included another siRNA to deplete REEP5 and reproduced the result on increased mitochondrial ROS levels in REEP5-depleted cells (revised Fig. 5B and C).

Minor points:

“1. In figure 1d, the cartoon of Mfn1/2 risks giving the wrong impression that the proteins have a similar structure or architecture to REEP5. Given that Mfn structure is known, the cartoon should more accurately reflect its domain structure. ”

We have modified the illustration to reflect the domain structures as suggested. The cartoon now resides in Fig. 1E in the revised manuscript.

Reviewer #2

“1. Co-IP between REEP/RTN and MFN were done mostly with overexpression. Notably, endogenous antibodies for both MFNs and several REEP/RTNs are commercially available. In endogenous co-IP shown in Fig. 1F, the setup is overly simplified. Whether MFN antibodies/beads have non-specific interactions with REEP5 is not controlled. Co-IP of other REEPs or RTNs was not compared. ”

We understand the concern about the potential caveat of using tagged expression in identifying protein interactions and have had conditions piloted for modulated expression levels to minimize potential artifacts. To experimentally address this issue, we tested endogenous protein interactions of MFN1/2 against a panel of REEP proteins and Reticulons. In the new Fig. 1C, MFN1 and 2 IP specifically enriched REEP5 but not the other proteins tested. Additionally, antibodies against MFN1/2 have no detectable cross-reactions with REEP5, reflected by our WB analysis of MFN1/2 antibodies and the commercial datasheet provided by the vendor (Fig. R1A and B).

Figure R1

Figure R1. (A) Immunoblotting analysis of endogenous MFN1/2 with antibodies from Cell Signaling Technology. The supposing position of endogenous REEP5 was indicated. (B) Data sheets of MFN1/2 antibodies provided by Cell Signaling Technology.

“2. In Fig. S1A and B, the authors showed that RTN3 had equivalent interactions with MFNs when compared to that of REEP5. However, when RTN3 was deleted in U2OS cells, the mitochondrial defects were not even close to that of REEP5. Could the authors comment on this discrepancy? Along this line, the KD efficiency in these cases were verified at neither protein nor transcript levels, and the depletion specificity was not verified by rescue experiments.”

In Fig. S1A, RTN3 was undetectable in the MFN1 pulldown. In Fig. S1B, we agree with the reviewer’s observation that RTN3 can be pulled down by MFN2, albeit weaker than REEP5. However, when endogenous protein interactions were analyzed by immunoprecipitation, only the REEP5-MFNs interactions were detectable. It appears that the affinity between endogenous REEP5 and MFN2 is much stronger than RTN3-MFN2, which is functionally reflected by the fact that Rtn3 depletion failed to give rise to mitochondrial collapse.

To ascertain that RTN3 does not contribute to mito-ER tethering, we used two additional anti-RTN3 siRNAs to mitigate potential off-target effects. The result (new Fig. S1C) again indicates that RTN3 knockdown yielded no detectable effect on mitochondrial morphology. This result suggests that the mitochondrial collapse is primarily caused by abolishing the REEP5-MFN1/2 tether. As suggested by the reviewer, siRNAs used to deplete REEP5 and RTN3 are verified at the protein level (new Fig. S1C top). The rescue experiment, as suggested, is carried out and the result is presented in the revised Fig. 4E.

“3. The selectivity of REEP5 by MFNs is striking and surprising, considering the sequence similarity between REEPs, or at least with REEP6. By truncation experiments, the authors showed that the C-terminal region of REEP5 is involved in direct interactions. It would thus make sense to construct a chimeric REEP with the tail of REEP5 replaced by that of REEP6. Such a chimera would be expected to maintain tubule-forming ability but lose mitochondrial contact. If so, the chimera could serve as a much better control in several tests, including mitochondrial positioning/movement and ROS regulation.”

Following this suggestion, two chimeric proteins were generated to further validate the specificity of the REEP5-MFNs interaction.

1. REEP5 N-terminus + REEP6 C-terminus and 2. REEP6 N-terminus + REEP5 C-terminus (revised Fig. S2B). Surprisingly, both chimeric proteins are able to interact with MFN2 (revised Fig. S2C). Given that full-length REEP6 does not show any interaction with MFN2, either in the ectopically expressed form or the endogenous form, it is likely that the N-terminal portion of REEP6 can mask or interfere with its C terminal domain, preventing its interaction with MFN1/2.

To further examine the selectivity of REEP5 by MFN1/2, we mutated REEP5 into a more REEP1-4-like sequence (Fig. S2D). We designed several point mutations intended to disrupt cAPH and CTH, the two domains in the C-terminus of REEP5 (revised Fig. S2D). Mutations in the CTH domain (REEP5 mut#1) affected the interaction substantially, whereas additional mutations in the cAPH domain (REEP5 mut#2) further abolished its interaction with MFN2 (revised Fig. S2E). Both mutants displayed normal ER patterns, indicating that they are able to maintain the ER-tubule forming ability (revised Fig. S2F).

We further tested the MFN2-binding-deficient REEP5 mutant (mut#2) in a number of settings with the following results:

- 1) Revised Fig. 4E and F, Wild type REEP5 rescued mitochondrial collapse, but mut#2 failed to do so.
- 2) Revised Fig. 4I, J and L, new Video 10 and 11, using doxycycline-inducible expression, wild-type REEP5 expression stimulated directional co-movement of mitochondria with ER. In contrast, mut#2 expression in parallel did not produce any increase in directional co-movement.
- 3) Mitochondrial distribution was highly correlated with ER as wild-type REEP5 accumulated (revised Fig. 4I), while this correlation decreased under REEP5 mut#2 overexpression, and completely abolished under REEP6 overexpression (revised Fig.

4J), further supporting the specificity of REEP5 in this ER-mitochondria correlation.

With respect to the ROS readout, the experimental setup cannot accommodate it since the fluorescent channels are fully occupied.

“4. The experiments shown in Fig. 1H is confusing and likely over-interpreted. FLIP experiments are usually informative in testing membrane continuity. In this case, the setup sounds more like a FRAP assay. Importantly, previous FRAP assays have shown that tubule-forming proteins are slower in mobility when compared to that of Sec61b. Therefore, the results could simply be read as their general mobility, but not their preference for contact. If the authors insist on this assay, they should compare the FRAP signal of REEP5 at the contact site in the presence or absence of MFNs, and Sec61b can now serve as a control for not sensitive to such changes. Finally, if the authors really meant for FRAP, the data should be analyzed as curves, but not representative images.”

We understand this comment. The reviewer is correct that the FLIP assay has been “repurposed” to mimic the FRAP assay with a different angle of interpretation. Recent study on organelle contact (PMID: 32179693), the region of interest (ROI) is repeatedly bleached over time to examine the behavior of the entire fluorescent pool. In our experiment, FLIP is performed to quench fluorescent signals surrounding the MERCs. If the fluorescent molecules are mobile and have access to the photobleached areas, the entire fluorescent pool will be diffused, as shown by the Sec61 β signal in the revised Fig. 2B and new Video 3. On the other hand, the tethering protein pairs remain co-immobilized and are thus spatially segregated.

Following the reviewer’s advice, we transfected MFN1/2 dKO MEFs follows: 1) REEP5 +/- MFN2; 2) Sec61b +/- MFN2 as a control. We found that REEP5 signals are largely dispersed in the absence of MFN2, but retained at the contact site when MFN2 is present. The Sec61b signal, other the other hand, is not sensitive to MFN2 status (revised Fig 2B and new Video 3). This new data appears to address the concern regarding the retarded mobility of tubule-forming proteins in general.

“5. Finally, several test conditions are considered very harsh to cells. For example, the overexpression of artificial ER-mito tether itself induces abnormality in mitochondrial morphology. The 1 mM H₂O₂ treatment should certainly cause a drastic fragmentation of mitochondria. Cautions should be taken when performing these assays or interpreting these results.”

We appreciate this important point. In our experiments, expression levels have been piloted to acquire sufficient visualization instead of maximum expression levels. As shown in revised Fig. S1D, the amount of ectopically expressed MFN2 is approximately 20% of the endogenous protein and is unlikely disruptive to the endogenous event. Therefore, the irreversible anchoring-induced mitochondrial hyperfusion morphology is more than likely a

biological effect, rather than arising from an overexpression artifact.

The H₂O₂ concentration we used (1 mM) was intended as a positive control for validating the mito-roGFP2 sensor cell line and no conclusions were drawn from H₂O₂-treated cells regarding the tethering. Nevertheless, we agree with the reviewer's concern regarding the harshness of such a dose. In the revised manuscript, we decreased the H₂O₂ concentration to 200 uM, and observed the same response with roGFP2 (revised Fig. 5A).

Reviewer #3

“1. It remains unclear whether *Mfn2* (or some of the many fusion constructs used in the study) localizes also to the ER under the conditions used in the manuscript. This was recently demonstrated in PMID: 34296790. It is therefore unclear where the described interaction takes place. Is it on the ER or on mitochondria?”

Martins et al suggested that MFN2 could be retrieved from the MAM via biochemical fractionation (PMID: 19052620). However, the amount of MFN2 is rather small (6.9%) and the REEP5-MFN2 tether may partially account for this observation. On the other hand, MFN1 is exquisitely localized to mitochondria. To address the reviewer’s comment, we performed three experiments to verify that the main interaction between REEP5 and MFN1/2 occurs *in trans* between ER and mitochondria.

1. Using the MFN1/2 dKO MEF cells, we observed that recruitment of free C-terminal fragment of REEP5 (which binds to MFN1/2) takes place on MFN2-positive, ER marker-negative structures (revised Fig. 2C and new Video 4). Furthermore, the cytosolic REEP5-C terminus caused the collapse of mitochondria by competitively inhibiting the endogenous REEP5-MFN2 interaction (revised Fig. 4G and S2G). These results are consistent with the notion that the interaction between REEP5 and MFN2 takes place between ER and mitochondria.
2. We constructed an MFN2 mutant with localized expression on the mitochondrial surface (MFN2^{ActA}) based on the publication cited above by the reviewer. IP analysis showed that REEP5 interacts with mitochondria-localized MFN2 (revised Fig. 1D). This result again supports the trans nature of the interaction between REEP5 and MFN2.
3. To further validate the tethering function of the REEP5 C-terminus, we replaced the cytoplasmic portion of LAMP1 with 3 copies of the REEP5 C-terminus. Expression of the chimera but not wildtype LAMP1 together with MFN2 indeed increased the contact frequency of mitochondria with the lysosome (revised Fig. 2D and E, and new Video 5), consistent with the idea that REEP5 and MFN2 form a tethering complex, which is consistent with our original finding that the endogenous REEP5 and MFN2 interact *in trans*.

“2. The authors claim that the *Mfn2*-REEP5 interaction regulates mitochondrial ROS levels. However, they cannot tell from their findings whether ROS were made within the mitochondria or elsewhere, e.g., the ER.”

This is an important point since protein folding in ER is also a prevalent source of ROS. To determine whether the increased ROS production originates from ER, we established a U2OS cell line stably expressing ER-Triper (revised Fig. S3A), a biosensor specific for ER ROS (PMID: 28347335). Quantification of the ROS imaging showed no change in ER ROS level (revised Fig. S3B and C). This result provided additional evidence that the increased

ROS is produced in the mitochondria.

3. *The sensitivity of the ROS probe used is insufficient. The HyPer7 is the gold standard for measuring mitochondrial ROS with high levels of accuracy and would be more suitable for the experiments performed here. More physiological treatments should be used to provide some information about the level of ROS increases or decreases described in the manuscript. Mfn2 KO or knockdown should also be measured in terms of ROS.*

We performed the following experiments as the reviewer suggested,

- 1) We established U2OS cells stably expressing mito-HyPer7, which responds to both H₂O₂ and DTT (revised Fig 5D). REEP5 KD in this cell line also significantly increased mitochondrial ROS level (revised Fig. 5E and F), which is consistent with our observation using roGFP2 in the original manuscript.
- 2) Similarly, we established MEF cells stably expressing mito-HyPer7, testing this cell line showed that MFN1/2 dKO leads to increased mitochondrial ROS (revised Fig. 5G and H).

Both results are included in the revised figures as specified above.

4. *Can the interaction be disrupted by expressing fragments of the postulated interaction domains?*

Yes. REEP5 interacts with MFN1/2 via its C-terminus. In 293T cells, REEP5 C-terminal over-expression competitively inhibited the endogenous REEP5-MFN2 interaction (revised Fig. S2G). Ectopically expressed REEP5 C-terminus in U2OS cells caused the collapse of mitochondria into peri-nuclear regions (revised Fig. 4G).

Minor Points:

1. *The co-immunoprecipitation of endogenous proteins is of unclear significance, since the relative input is not defined.*

We have updated the figure annotation to indicate that 0.5% of whole cell lysate was used as input.

2. *The mitochondria-ER contacts are defined with an unusual abbreviation. Most commonly, these are referred to as MERCs (or biochemically as MAMs).*

We have updated the text accordingly.

April 12, 2024

Re: JCB manuscript #202304031R

Dr. Zhi Hong
Department of Neurology
Second Affiliated Hospital of Zhejiang University School of Medicine
866 Yuhangtang Rd
Hangzhou 310058
China

Dear Dr. Hong,

Thank you for submitting your revised manuscript entitled "Dynamic interaction of REEP5-MFN1/2 enables mitochondrial hitchhiking on tubular ER." Thank you for your patience with the review process. The manuscript has been seen by two of the original reviewers whose comments are appended below. While the reviewers are overall positive about the work in terms of its suitability for JCB, some important issues remain.

You will see that Reviewer #1 requests additional method details and one more quantification of existing data. Reviewer #2 asks to tone down the claim that REEP6 does not interact with MFNs and to explain the different sizes of the MFNs bands in IP versus whole cell lysate samples. Looking at the Source Data images, it does seem that the MFN1 and MFN2 bands in IP fractions are consistently running at a higher molecular weight than in the WCL samples. This is most clearly visible in the Source Data for Figure S2C. It may be that REEP5 binds to a subpopulation of MFNs with some posttranslational modifications and this difference in sizes as well as other possible explanations should be discussed in the text. In some figures, such as S2G, the IP and WCL panels are not aligned accurately giving the impression that in this experiment all the MFN bands ran at the same size. It is essential for you to carefully realign all figures that have MFN blots in order to show the correct relationships between IP and WCL fractions.

Our general policy is that papers are considered through only one revision cycle; however, given that the suggested changes are relatively minor we are open to one additional short round of revision. Please note that we will expect to make a final decision without additional reviewer input upon resubmission.

Please submit the final revision within one month, along with a cover letter that includes a point by point response to the remaining reviewer comments.

While revising the figures please try to avoid pairing red and green colors for images and graphs to ensure legibility for color-blind readers. If red and green are paired for images, please ensure that the particular red and green hues used in micrographs are distinctive with any of the colorblind types, such as magenta/green. If not, please modify colors accordingly or provide separate images of the individual channels.

Thank you for this interesting contribution to Journal of Cell Biology. You can contact the journal office with any questions at cellbio@rockefeller.edu.

Sincerely,

Laura Lackner, PhD
Monitoring Editor
Journal of Cell Biology

Dan Simon, PhD
Scientific Editor
Journal of Cell Biology

Reviewer #1 (Comments to the Authors (Required)):

The revision largely addresses my concerns. There are a few important missing pieces of information. 1) In Fig. 2B, the number of cells from which the FLIP results were obtained needs to be indicated. 2) In Fig. 4D&F, a plot quantifying the distance from nucleus out is shown but there is no explanation for how the plot was done. Was it a line plot or total fluorescence? 3) For Fig. 4G needs quantification. Why was a similar plot to D&F not done?

Reviewer #2 (Comments to the Authors (Required)):

In this revision, the authors have addressed most of my concerns. I do however have some comments for the authors to consider. Regarding the endogenous co-IP shown in Figure 1C, the REEP5 bands are not well aligned. The band in MFN2 IP appears to be higher than what is shown in the input. If so, there are some signals in the IgG lane. Is that a shadow of a non-specific band, or in fact REEP5? Both MFN1 and MFN2 appear to be higher in the IP lanes when compared to the input lanes. Why? Some RTN bands are too close to the edge of the panel. I could have questioned the confidence of the results, but the situation in Figure S2G are better in these regards. In short, the same quality is preferred for Figure 1C. Regarding the REEP5 specificity, I am not entirely convinced that only REEP5 interacts with MFN. At least REEP6 remains suspicious, given the tail swapping experiments. The fact that endogenous REEP6 does not interact much with MFNs could simply be that REEP6 is not abundant at all in the tested cells. See how much REEP5 is present in the WCL and how little it is pulled down. The mutations (#1 and #2) represent too much alteration of the sequence. In any case, it is recommended that the authors tune down the conclusion on this point.

Zhi Hong, PhD
Assistant Professor
Zhejiang University–University of
Edinburgh Institute
Building 2A, Room A215,
718 East Haizhou Rd., Haining,
Zhejiang, P.R. China
Phone: +86- 19558420607
Email: zhihong@intl.zju.edu.cn

May 1, 2024

Laura Lackner, PhD
Monitoring Editor
Journal of Cell Biology

Dan Simon, PhD
Scientific Editor
Journal of Cell Biology

Dear Laura and Dan,

Please find attached our revised manuscript **“Dynamic interaction of REEP5-MFN1/2 enables mitochondrial hitchhiking on tubular ER”** (Manuscript number: #202304031). I am grateful to your interest in our work. This whole revision process has greatly improved this paper.

A point-by-point reply to the reviewers' comments goes as following:

Reviewer #1

“The revision largely addresses my concerns. There are a few important missing pieces of information.”

“1) In Fig. 2B, the number of cells from which the FLIP results were obtained needs to be indicated.”

In Fig. 2B, individual cells were imaged for each group from 3 independent experiments. Representative images for each group are shown. We have added this information to the legend of Fig. 2B.

“2) In Fig. 4D&F, a plot quantifying the distance from nucleus out is shown but there is no explanation for how the plot was done. Was it a line plot or total fluorescence?”

We followed the standard “Clock Scan” (Image J plug-in) protocol to generate the mean intensity of mitochondrial signal as a function of radial distance to the center of mitochondrial network. We added the detailed description of quantification in the method part, sentences highlighted in yellow (pg. 18, line 456-464).

“3) For Fig. 4G needs quantification. Why was a similar plot to D&F not done?”

We have added the quantification data as new Fig. 4H in our revised manuscript.

Reviewer #2

“In this revision, the authors have addressed most of my concerns. I do however have some comments for the authors to consider.”

“Regarding the endogenous co-IP shown in Figure 1C, the REEP5 bands are not well aligned.

The band in MFN2 IP appears to be higher than what is shown in the input. If so, there are some signals in the IgG lane. Is that a shadow of a non-specific band, or in fact REEP5? Both MFN1 and MFN2 appear to be higher in the IP lanes when compared to the input lanes. Why? Some RTN bands are too close to the edge of the panel. I could have questioned the confidence of the results, but the situation in Figure S2G are better in these regards. In short, the same quality is preferred for Figure 1C.”

Following this suggestion, we increased the washing stringency of the endogenous IP and lowered the non-specific bands in the IgG control lane. As shown in the revised Fig. 1C and its source data, the REEP5 band in the MFN2 IP lane improved significantly.

The shifted mobilities of MFN1/2 between IP and the input lanes were consistently observed. The shift becomes even more obvious when gels were run for a longer time as in Fig. 1C. It is possible that REEP5 binds with a subpopulation of posttranslationally modified MFN1/2, which is an interesting topic for us to pursue in a future study. We have updated the text in the Discussion (pg. 12, line 308-311).

“Regarding the REEP5 specificity, I am not entirely convinced that only REEP5 interacts with MFN. At least REEP6 remains suspicious, given the tail swapping experiments. The fact that endogenous REEP6 does not interact much with MFNs could simply be that REEP6 is not abundant at all in the tested cells. See how much REEP5 is present in the WCL and how little it is pulled down. The mutations (#1 and #2) represent too much alteration of the sequence. In any case, it is recommended that the authors tune down the conclusion on this point.”

I agree with the reviewer that we cannot exclude the possibility that MFN1/2 interact with the other REEPs/RTNs in certain scenarios. We have retuned our conclusion, by stating that “endogenous MFN1/2 consistently interacts with REEP5”, instead of using “specifically”.

In addition, we have switched the red color into magenta to accommodate color-blind readers.

We hope that you find the final revision suitable for publication in JCB. We thank you and the reviewers for the positive feedback of this work.

Sincerely,

Zhi Hong, Ph.D.